# Light in, sound keys out: photoacoustic PUFs from stochastic nanocomposites

Taehyun Park [1,2,6], Junhyung Kim [3,6], Raksan Ko[1,6], Byullee Park[3,4,5] ✉ & Hocheon Yoo [2] ✉

We present a concept of physically unclonable functions utilizing the photoacoustic effect to generate structurally random, inference-resistant cryptographic keys. The system consists of a $CuO/SnO_2$ nanoparticle composite, where CuO acts as a visible-range absorber and $SnO_2$ serves as a non-absorbing dispersive matrix. Nanosecond laser pulses induce localized heating and acoustic wave emission, providing spatially heterogeneous photoacoustic signals that are digitized into binary matrices. Evaluations across ten devices yielded a bit uniformity of 49.54%, inter-device Hamming distance of 49.69%, entropy of 0.983, and bit aliasing of 49.38%—all approaching ideal values for secure key generation. Machine learning attacks using logistic regression and support vector machines failed to infer underlying patterns, with prediction accuracies of 53.53% and 52.54%. The device maintains cryptographic performance after transfer to diverse substrates, including human skin, highlighting its mechanical adaptability. This subsurface, light-to-sound-based approach offers a scalable platform for secure authentication on flexible or opaque surfaces.

As digital security threats continue to evolve, the need for robust hardware security solutions grows more urgent[1,2]. Physically unclonable functions (PUFs) provide a foundation for device authentication by using inherent physical randomness to generate unique, tamper-resistant cryptographic keys[3]. Early PUF designs relied on electronic mechanisms, such as delay variations in ring oscillators[4] and transistor mismatches[5] in SRAM cells. While effective, these methods often suffered from environmental sensitivity and vulnerability to machine learning-based attacks. Furthermore, as CMOS-based circuits, they required complex fabrication processes and intricate circuit designs, making low-cost production challenging. Their reliance on rigid wafer-based substrates also imposed limitations, restricting applications in wearable[6], flexible[7], and embedded security systems[8] that demand adaptability beyond conventional silicon platforms. Recent shifts toward adaptive, physically interactive hardware platforms further highlight this limitation[9].

Beyond conventional PUFs, alternative classes of material-based PUFs have emerged, offering alternative approaches that move away from CMOS fabrication constraints. Rather than silicon-based designs that rely on transistor-level mismatches or precisely controlled lithographic processes, material-based PUFs exploit the intrinsic variations within materials themselves. These approaches introduce randomness through controlled material heterogeneity, such as blending different substances[10], inducing localized doping effects[11], embedding nanoparticles[12], or grain boundary distributions[13]. By shifting the source of entropy from circuit-level variations to material-level randomness, these methods simplify the overall fabrication process while maintaining strong security properties. For example, blending two materials can introduce significant stochastic variations through simple deposition techniques such as spin-coating[14]. Similarly, block copolymer self-assembly has been shown to produce reproducible yet

[1]Department of Semiconductor Engineering, Gachon University, Seongnam, Republic of Korea. [2]Department of Electronic Engineering, Hanyang University, Seoul, Republic of Korea. [3]Department of MetaBioHealth, Sungkyunkwan University, Suwon, Republic of Korea. [4]Department of Biophysics, Institute of Quantum Biophysics, Sungkyunkwan University, Suwon, Republic of Korea. [5]Department of Biopharmaceutical Convergence, Sungkyunkwan University, Suwon, Republic of Korea. [6]These authors contributed equally: Taehyun Park, Junhyung Kim, Raksan Ko. ✉e-mail: byullee@skku.edu; hocheon@hanyang.ac.kr

spatially diverse nanoscale patterns, offering a scalable route to encode physical randomness at the material level[7,15]. Different from CMOS PUFs that require multi-step photolithography and precise doping control, this approach enables large-scale randomness formation in a single step. These advancements make material-based PUFs particularly attractive for low-cost, flexible, and scalable security solutions, expanding their potential applications in embedded and wearable systems where conventional silicon PUFs face structural and manufacturing limitations.

Yet, despite the advantages of material-based electronic PUFs, several limitations remain. (i) First, the number of physically generated random bits scales with the number of individual devices, meaning that achieving large-scale randomness requires a proportional increase in the number of physical devices. This presents a critical challenge in array-based architectures, where increasing the number of devices also demands a corresponding increase in current signal levels. However, in large-scale arrays, maintaining sufficient signal contrast across all devices becomes difficult, ultimately limiting scalability in high-density PUF implementations. (ii) Second, electronic PUFs require electrode patterns to read out electrical properties, which introduces additional design constraints. The presence of readout electrodes not only increases chip size but also restricts the achievable spatial resolution of material-level variations. While materials themselves may exhibit fine-grained randomness at the nanoscale, the resolution of randomness extraction is ultimately dictated by the electrode layout, making it difficult to fully utilize high-resolution material heterogeneity. Additionally, the need for well-defined electrode interfaces prevents further chip miniaturization, as shrinking the device size would also require downsizing the electrode pitch, which is not always feasible in standard fabrication processes. These challenges highlight the structural and scalability bottlenecks of existing material-based electronic PUFs, motivating the exploration of alternative architectures that can overcome these limitations.

In this work, to address these limitations, we introduce a photoacoustic (PA)-PUF, which departs from electronic readout mechanisms and instead utilizes ultrasound signals (i.e., PA signals)[16,17] generated by optically excited nanoparticles. While the conventional approaches of PUFs previously reported require a large number of addressable elements and electrode-based signal extraction, the PA-PUF captures spatially distributed subsurface randomness without the need for predefined circuit patterns. By embedding a composite of CuO (photoabsorbing) and SnO$_2$ (dispersive) nanoparticles (NPs) into a solid matrix, this system forms a stochastic material structure where optical excitation induces localized thermal expansion, producing unique PA responses[18]. These responses translate into high-entropy cryptographic keys, extracted without reliance on direct electrical measurements. During acquisition, each response pattern was obtained at a rate of ~10$^4$ pixels per second across a $3 \times 3 \, mm^2$ area. Experimentally obtained results confirm that the PA-PUF generates robust and high-quality random bit sequences, achieving near-ideal uniformity (49.54%), inter-device Hamming distance (49.69%), and entropy (0.983). Furthermore, the absence of electrical contacts enables flexible and transferable implementations, demonstrated by successful integration onto curved and wearable substrates, including human skin. Security evaluations show that machine learning attacks fail to infer PA-PUF responses, reinforcing the system's resilience against AI-driven threats.

## Results

The use of the PA effect as the operating principle for a PUF marks a conceptual shift from conventional optical approaches. As shown in Fig. 1, this architecture employs nanosecond-pulsed laser excitation to induce localized thermal expansion in light-absorbing NPs, leading to the emission of PA waves[19,20]. These acoustic signals are then captured and reconstructed into spatially heterogeneous intensity maps[21–23],

which are binarized to generate a digital cryptographic key. Unlike conventional optical PUFs, which rely on surface morphology-based transmission phenomena[24,25], this method captures subsurface randomness that originates from the spatial distribution of optically responsive NPs embedded within an inert matrix. In our design, CuO NPs—absorbers in the visible range—serve as the primary contributors to PA signal generation, while SnO$_2$ NPs function as dispersive elements without contributing to optical absorption. This binary composition is intended to modulate local absorption on the nanoscale, disrupt spatial regularity, and introduce stochastic variations in acoustic signal intensity. This intrinsic randomness enhances unpredictability and strengthens resistance to replication. Rather than relying on transmission-based optical systems that require transparent substrates, the PA-PUF operates in reflection mode, enabling its application on opaque surfaces. Furthermore, the dual dependence on light absorption and ultrasound transduction adds structural complexity that complicates replication via lithographic or imprinting techniques[26].

To validate the hypothesis that material-level heterogeneity contributes to subsurface acoustic randomness, we first investigated the morphological and conceptual foundations of the CuO/SnO$_2$ composite film. Optical micrographs in Fig. 2a, captured using both digital microscopy and bright-field microscopy, display that the composite film exhibits a randomly distributed yet uniformly covered surface. No visible phase boundaries or aggregation patterns were observed, suggesting a homogenous intermixing of the two NP species across the film area. Figure 2b provides a schematic illustration that compares the expected structure of single-component and mixed NP films. In CuO-only configurations, large particle size and surface energy promote aggregation, whereas SnO$_2$-only films form a dense, uniform layer due to their nanoscale dimensions. In the mixed architecture, SnO$_2$ NPs effectively infiltrate the voids between CuO NPs, suppressing their tendency to cluster and enhancing uniformity in terms of space. To confirm that CuO and SnO$_2$ NPs were synthesized with the intended composition and oxidation states, we performed X-ray photoelectron spectroscopy (XPS) analyses on each material before forming the composite. Independent validation of their physical and chemical identities is essential, as the composite design relies on CuO functioning as the optically absorbing component, while SnO$_2$ serves as a dispersive matrix without optical interference. As shown in Fig. 2c and Supplementary Fig. 1, the Cu $2p_3/_2$ and Cu $2p_1/_2$ peaks appeared at 933.6 and 953.5 eV, accompanied by characteristic satellite features between 943 and 945 eV, confirming the Cu$^{2+}$ oxidation state and formation of CuO[27]. In the case of SnO$_2$ film, the XPS spectra exhibited Sn $3d_5/_2$ and $3d_3/_2$ peaks at 487.2 eV and 495.6 eV, respectively, corresponding to Sn$^{4+}$, indicating that the synthesized particles possess the oxidation state of SnO$_2$[28]. These results validate the chemical integrity of both oxide systems prior to mixing. Following this verification, we analyzed the particle size characteristics of the individual and mixed NP suspensions using dynamic light scattering (DLS) analysis (Fig. 2d). CuO particles exhibited a mean hydrodynamic diameter of 208.5 nm, while SnO$_2$ NPs measured 18.6 nm. The mixed solution maintained a bimodal distribution, indicating that both NP species remained physically distinct without aggregation or fusion. This supports the hypothesis that SnO$_2$ functions as a physical spacer between larger CuO particles, enhancing dispersion and uniformity within the composite.

Building on the particle size characteristics confirmed by DLS, we next investigated the surface morphology of the films to understand how structural dispersion manifests at the microscale. While optical microscopy results in Fig. 2a confirmed uniform surface coverage, more detailed topographical insights were obtained through scanning electron microscopy (SEM) and energy-dispersive X-ray spectroscopy (EDS) analyses (Fig. 2e). Compared to the CuO and SnO$_2$ single-component films (Supplementary Figs. 2 and 3), the composite film

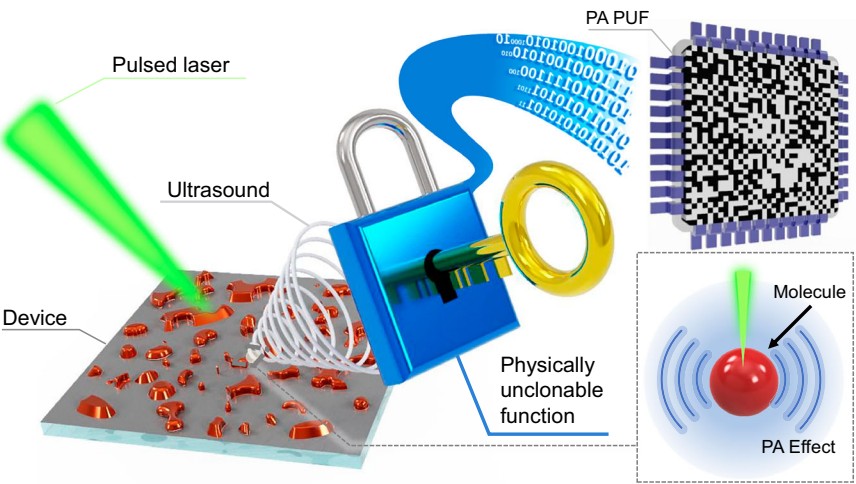

**Fig. 1 | Conceptual diagram of the photoacoustic (PA) effect-based physically unclonable function (PUF) framework.** A pulsed laser irradiates the nanostructured device, generating PA signals through the PA effect. The resulting signals are converted into a binary PUF response for hardware security application.

exhibited a more homogeneous surface with no signs of large-scale aggregation, suggesting effective intermixing between the two NP species. Elemental mapping through EDS further supported this observation: in each individual film, the spatial distribution of Cu and Sn signals reflected their respective morphological features, reinforcing the correlation between particle characteristics and elemental presence. In the composite film, both Cu and Sn signals appeared uniformly across the scanned area, indicating that the two types of NPs were not phase-separated but well-dispersed throughout the matrix. Atomic force microscopy measurements (Fig. 2f) provided further evidence of nanoscale dispersion. CuO-only samples displayed uneven surface features due to particle agglomeration, while $SnO_2$ films appeared smooth and continuous. In contrast, the composite film showed moderate roughness and finely textured surfaces, indicative of spatial separation between CuO domains and surrounding $SnO_2$ particles. To quantitatively evaluate the thickness uniformity of the CuO/$SnO_2$ composite film across the full deposition area ($2.5 \times 2.5$ cm²), we conducted profilometry and cross-sectional SEM measurements over nine representative regions distributed throughout the substrate (Supplementary Fig. 4). Line-scan profilometry revealed consistent height profiles over 100 μm distances with no abrupt changes. The median thickness values extracted from each region ranged from 129.3 to 152.1 nm, with an average of 140.31 nm and a standard deviation of ±10.05 nm. This corresponds to less than ±10% deviation from the mean thickness, indicating high spatial uniformity across the film. Cross-sectional SEM images obtained at the same regions (Supplementary Fig. 5) independently confirmed this uniformity, with measured thicknesses closely matching the profilometry results.

UV–vis absorption spectra (Fig. 2g) revealed distinct optical behaviors for each material. The CuO NPs film exhibited broadband absorption across the visible spectrum (400–700 nm)[29,30], with absorbance increasing at shorter wavelengths. In contrast, the $SnO_2$ NPs film exhibited no considerable absorption within the measured wavelength range, consistent with its wide energy bandgap. The composite film followed the spectral trend of CuO, suggesting that the PA signal originates primarily from the CuO component. Tauc plot analysis (Supplementary Fig. 6) revealed both direct and indirect optical transitions in CuO (1.64 eV and 1.24 eV, respectively), while $SnO_2$ exhibited a large direct bandgap of 4.29 eV and an indirect transition at 3.21 eV. The photon energy used for PA excitation (532 nm, 2.33 eV) exceeds both band gaps of CuO, allowing absorption through both direct and phonon-assisted indirect transitions. In contrast, the excitation energy is insufficient for electronic transitions in $SnO_2$, indicating that $SnO_2$ serves primarily as a spatial randomizing

matrix rather than a PA-active component. Above-described experimental results confirm that the CuO/$SnO_2$ composite integrates chemically and structurally distinct species that retain their respective functions—light absorption by CuO and spatial dispersion by $SnO_2$—creating a heterogeneous matrix capable of producing stochastic subsurface PA responses.

To investigate whether the structural randomness of the CuO/$SnO_2$ composite film leads to functional diversity in PA responses, we conducted PA imaging on three NP film types: CuO-only, $SnO_2$-only, and the CuO/$SnO_2$ composite. The PA-PUF imaging performance was evaluated using a reflection-mode optical-resolution photoacoustic microscope (OR-PAM) system, as illustrated in Fig. 3a. This system enables high-resolution and wide-field imaging through a hybrid scanning mechanism. A fast one-axis galvanometer (GM) scanner performed rapid angular scanning along the x-axis, while motorized translation stages enabled precise movement along the y-axis, facilitating raster scanning. Figure 3b provides a closer view of the imaging head, where the optical and acoustic components are integrated. The focused beam passes through the ring-type ultrasound transducer (RUT), which enables coaxial optical illumination and acoustic detection. This design optimizes PA signal acquisition while maintaining a clear optical path. The beam is directed via a prism mirror (PM) and focused onto the sample using the objective lens (OL), ensuring precise light delivery to the target. The GM scanner controls the angular deflection of the beam, which is critical for fast B-scan acquisition. The maximum achievable B-scan rate of our original system is 250 Hz, but in this study, a B-scan rate of 10 Hz was used due to the 20 kHz pulse repetition frequency (PRF) limitation of the laser[31]. For image acquisition, a scan range of 3 mm with 1000 lateral pixels was employed, corresponding to a pixel size of 3 μm. Given these settings, 1000 B-scans were acquired in 100 s[32], resulting in a final PA MAP image of 1000 × 1000 pixels. The lateral resolution of the system was quantified and are shown in Fig. 3c. The lateral resolution was evaluated using the edge spread function of a razor blade, yielding a full-width at half maximum value of 10 μm (Fig. 3c). This value closely matched the theoretical prediction (Supplementary Note 1). This measurement approach enables localized response capturing from different regions within the same film, eliminating the need to fabricate distinct, individually addressable PUF units. While the spatial resolution of the OR-PAM system (~10 μm) exceeds the size of individual CuO and $SnO_2$ nanoparticles (<100 nm), the primary source of randomness is not individual particle contrast, but the emergent mesoscale aggregation patterns formed during film deposition (Supplementary Fig. 7) These stochastic structures are highly sensitive to fabrication conditions and

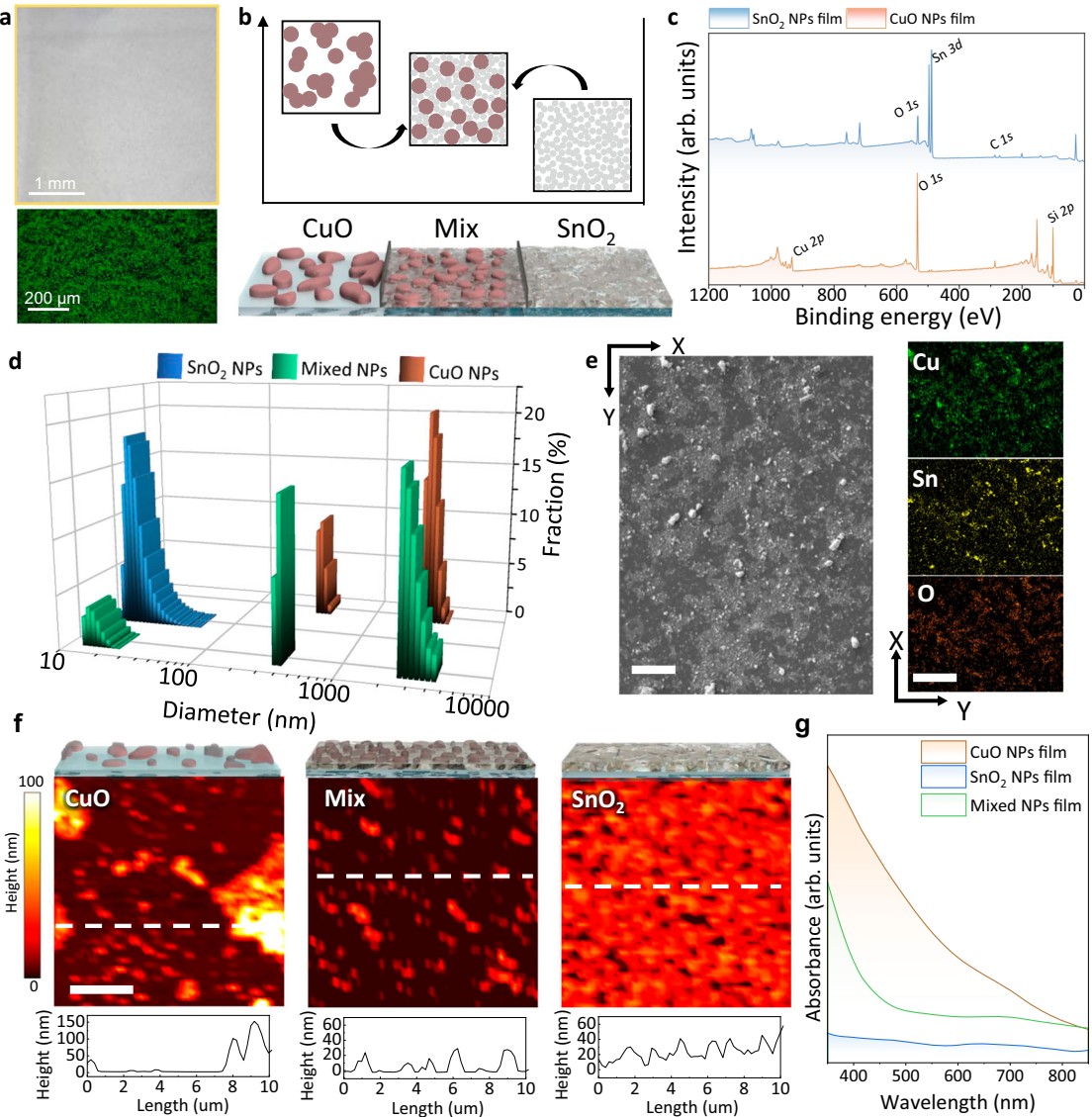

**Fig. 2 | Structural, morphological, and optical characterization of the mixed nanoparticle (NP) films for photoacoustic physically unclonable function (PA-PUF) implementation. a** Digital microscope image (top) and bright-field microscope image (bottom) of the fabricated mixed NP film, showing surface distribution of CuO and SnO$_2$ NPs. **b** Schematic diagram comparing the mixed NP configuration with individual CuO and SnO$_2$ NP types for structural design. **c** X-ray photoemission spectroscopy spectra of CuO and SnO$_2$ NPs films confirming the formation of CuO and SnO$_2$, respectively. **d** Dynamic light scattering analysis results of CuO, SnO$_2$, and mixed NP compositions. Average particle sizes are 208.5 nm for CuO and 18.6 nm for SnO$_2$. **e** Scanning electron microscopic image and energy dispersive X-ray spectroscopy analysis of the mixed NP surface, showing randomly and uniformly distributed CuO and SnO$_2$ NPs. The scale bar represents 100 μm. **f** atomic force microscopy analysis of CuO, SnO$_2$, and mixed NP films, comparing surface morphology and dispersion characteristics. The scale bar represents 3 μm. **g** UV–Vis absorbance spectra of CuO, SnO$_2$, and mixed NP films, showing combined optical properties relevant to PA-effect-based PUF operation.

effectively irreproducible, providing a robust entropy source for the PUF. By decoupling bit generation from discrete device count, this configuration addresses a key limitation in array-based systems, where expanding random bit output often requires a proportional increase in device complexity.

Supplementary Figs. 8 and 9 display representative PA signal maps for each film. The CuO/SnO$_2$ film generated a heterogeneous acoustic intensity map without noticeable periodicity or spatial bias. In contrast, the CuO-only film produced spatially clustered signal hotspots, attributed to particle agglomeration. As expected, the SnO$_2$-only film showed no detectable PA signal under 532 nm excitation due to its wide bandgap and lack of optical absorption (Supplementary Fig. 9). To convert these PA responses into digital cryptographic keys, each collected grayscale image (1000 × 1000 pixels) was denoised using a bandpass filter and then downsampled via binning. Specifically,

neighboring 4 × 4 pixel blocks were averaged and thresholded to generate binary values, resulting in a 250 × 250 bit matrix for each PA MAP image (Fig. 3d). A total of ten PA-PUF responses were extracted from locationally random, non-overlapping regions of the same CuO/SnO$_2$ film. The cryptographic quality of the resulting PUFs was evaluated using five standard metrics: uniformity, inter-HD, entropy, bit aliasing, and intra-device repeatability.

The uniformity, defined as the fraction of "1" bits in a response, exhibited an average mean value of 49.54% across ten devices, with an average standard deviation of 7.29% (Fig. 3e). This near-ideal balance confirms unbiased bit generation with no systemic preference for either binary state. The inter-HD values, which quantify the bit-level dissimilarity between all device pairs, were calculated to average value of 49.69% with average standard deviation of 5.73% (Fig. 3f), closely approximating the theoretical maximum of 50%. This demonstrates

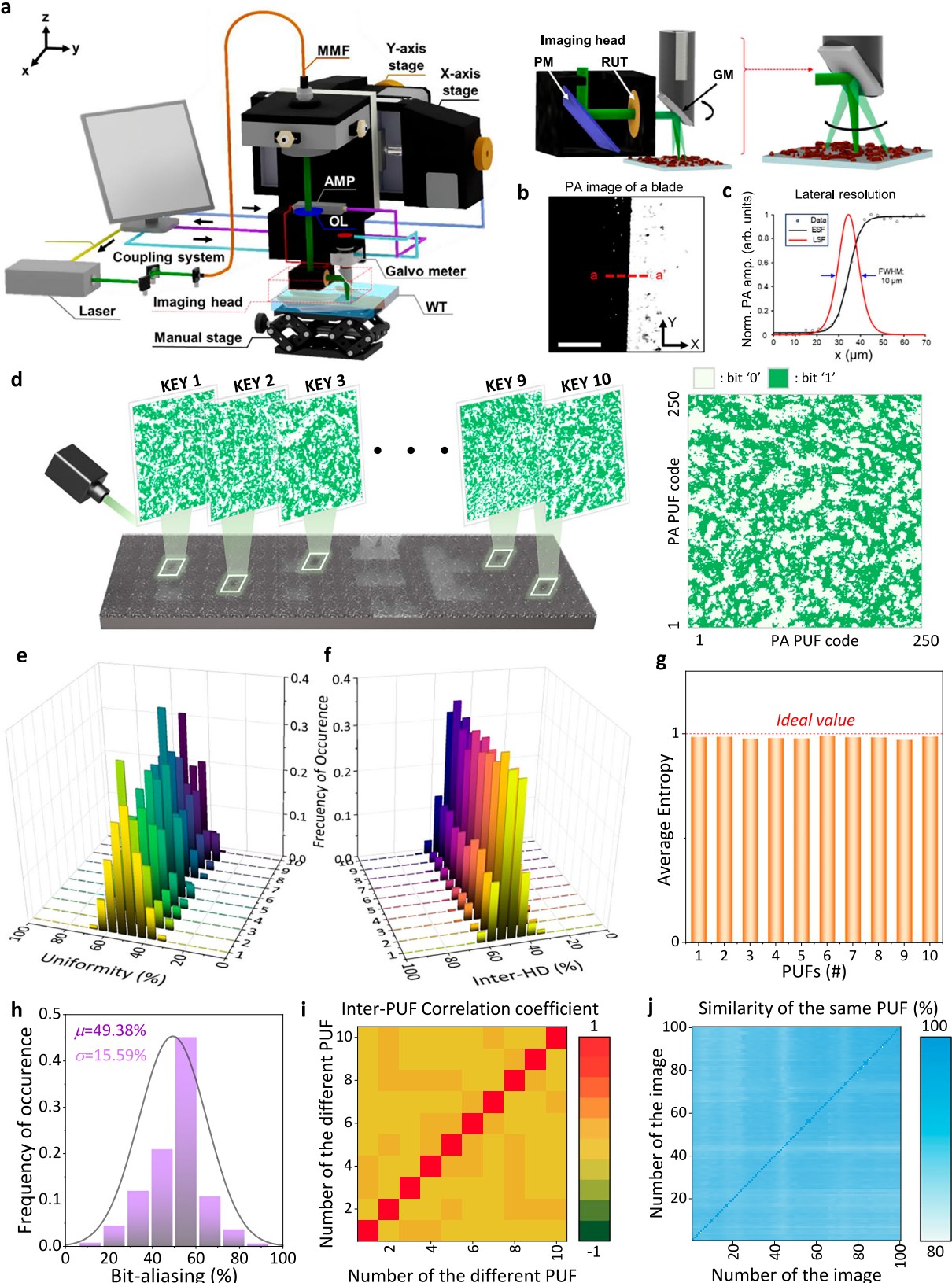

that each PUF instance is well-separated in the binary space, a key requirement for preventing pattern overlaps. As displayed in Fig. 3g and Supplementary Fig. 10, the average entropy value approached 0.983, which is close to the ideal case of 1, suggesting high randomness and minimal redundancy. Bit aliasing (Fig. 3h), calculated as the average probability of a "1" at each bit position across the ten devices, was centered at 49.38%, indicating a balanced and unbiased spatial distribution of bits. The ability to derive cryptographic responses from multiple non-overlapping regions within the same film suggests a level of scalability that does not rely on segmented circuitry or individually configured PUF cells. Instead, randomness emerges directly from the intrinsic heterogeneity of the nanocomposite, allowing high-entropy

**Fig. 3 | Optical-resolution photoacoustic microscope (OR-PAM) system and statistical evaluation of PA-PUF performance. a** Schematic illustration of the OR-PAM system for generating PA-PUFs. **b** Magnified view of the imaging head and raster scanning scheme using galvanometer (GM). **c** PA image of a razor blade and corresponding lateral resolution measurement based on the line spread function; the full width at half maximum was measured to be 10 μm. **d** Schematic diagram of a PA PUF array fabricated on a single plane, with each pattern consisting of $250 \times 250$ bits. **e** Uniformity distribution of PA PUF responses extracted from 10 different positions. **f** Inter-hamming distance (inter-HD) distribution of PA PUF patterns from 10 spatially separated regions. **g** Entropy analysis of PA PUF responses. **h** Bit-aliasing values calculated from 10 PA PUF devices. **i** Inter-PUF correlation matrix showing pairwise similarity between different devices. **j** Intra-PUF stability analysis over 100 repeated measurements from a single device.

key extraction without physical layout constraints. This decoupling of randomness generation from device-level scaling offers a practical advantage in systems where spatial compactness and fabrication simplicity are critical. These observations, in addition to demonstrating structural advantages, are consistent with our initial hypothesis regarding the function of each component within the composite structure. The CuO-only PA PUF, while capable of generating PA signals, exhibited significant aggregation-induced clustering, which led to a significant degradation in cryptographic metrics. Specifically, their uniformity, inter-HD, and entropy averaged 9.75%, 17.56%, and 0.455, respectively (Supplementary Fig. 11), substantially deviating from the ideal 50% and 1 benchmarks. This imbalance indicates systemic bias and reduced key separability. By contrast, the $SnO_2$-only PA PUF generated negligible acoustic responses due to insufficient optical absorption, resulting in nearly blank binary outputs with low values approaching zero at each figure of merit (Supplementary Fig. 12). These results underscore the inadequacy of single-component systems for PUF applications.

The above random metrics were achieved by optimizing both the $CuO: SnO_2$ mass ratio and the total NP concentration, based on the interplay between PA signal strength and spatial dispersion. As represented in Supplementary Fig. 13, at a fixed total concentration of 60 mg·mL$^{-1}$, films were prepared with mass ratios of 2:1, 1:1, and 1:2. Increasing the CuO content enhanced signal intensity but also led to pixel saturation and an overrepresentation of "1" bits in the binarized output. In contrast, increasing the $SnO_2$ fraction diluted the PA-active component, resulting in weaker signals and a bias toward "0". Only the 1:1 ratio maintained a sufficient PA response while ensuring uniform spatial randomness, yielding near-ideal values in uniformity, inter-HD, and entropy. A separate concentration sweep (20–180 mg·mL$^{-1}$) performed at the 1:1 ratio further confirmed that signal intensity must remain within an optimal range to preserve response quality. At low concentrations, the reduced number of absorbers lowered PA amplitude, leading to sparsely populated bitstreams. At high concentrations, signal saturation introduced strong local contrast and clustering artifacts. The intermediate value of 60 mg·mL$^{-1}$ achieved the best balance, producing stable bit distributions with minimal aggregation-induced bias. These observations confirm that both PA signal generation and randomness characteristics are strongly governed by material formulation (Supplementary Fig. 14).

Beyond confirming the overall randomness and diversity of individual patterns, further analyses were conducted to investigate whether the hybrid PA-PUF design also maintains statistical independence between devices and temporal stability within each device. As represented in Fig. 3i, the inter-PUF correlation value showed negligible correlation between any two devices, with all off-diagonal elements remaining near zero. This statistical independence among devices further supports the conclusion that the PA-PUF responses are intrinsically unique. Finally, to evaluate temporal stability, the same PUF area was measured 100 times under identical conditions. The resulting intra-HD indicated an average bit-level agreement of 91% (Fig. 3j), reflecting considerable repeatability and robustness. These statistical outcomes validate our material design strategy: the $CuO/SnO_2$ nanocomposite enables physically random, optically activated PA responses that are robust, cryptographically secure, and spatially extensible. To further evaluate the practical robustness of the PA-PUF key generation process, we performed additional sensitivity analyses to examine the effect of minor variations in preprocessing conditions. Specifically, we assessed how small shifts in binarization thresholds (±5%, ±10%) and changes in binning sizes ($4 \times 4$, $8 \times 8$, $16 \times 16$) affect the extracted bitstream quality. For each condition, we recalculated key figures of merit, including inter-HD, entropy, and uniformity. As shown in Supplementary Figs. 15 and 16, all metrics remained stable across the tested ranges. For instance, under threshold variation (Supplementary Fig. 15), the 95% confidence intervals for inter-HD, entropy, and uniformity were $49.25 \pm 0.08\%$, $0.9916 \pm 0.0013$, and $50.02 \pm 1.93\%$, respectively. Similarly, under bin size variation (Supplementary Fig. 16), these values were $48.92 \pm 1.31$, $0.9855 \pm 0.0098$, and $49.98 \pm 0.028\%$, indicating minimal degradation. These results suggest that the PA-PUF are resilient against minor perturbations in processing parameters, demonstrating the robustness and reliability of the system.

While the previous analyses confirm the intrinsic randomness and stability of the PA-PUF structure, practical implementation requires that this performance be maintained when the device is deployed across a variety of real-world substrates[33–35]. To this end, we developed a transferable, stamp-like PA-PUF architecture composed of a CuO/$SnO_2$ NP layer encapsulated within a multilayer structure including parylene films (Fig. 4a). This configuration allows the PUF device to be physically applied to both rigid and soft surfaces without structural degradation. The stamp-like PA-PUF was successfully transferred onto multiple types of surfaces. As shown in Fig. 4b, the device conformed smoothly to a glass bottle, demonstrating its applicability to curved, non-biological platforms. Figure 4c further illustrates the device applied to human skin, representing a practical and challenging real-world scenario for wearable authentication. To evaluate whether the cryptographic patterns of the PA-PUF remain secure under such platform transitions, we performed a machine learning–based inference test using the sequence illustrated in Fig. 4d. A single PA-PUF response, visualized in Fig. 4e, was measured by in vivo human hand attached condition and sequentially digitized to the form of a $250 \times 250$ binary matrix. Prior to the machine learning based test, the randomness of the in vivo-type PA-PUF was quantified using the previously introduced figures of merit. As illustrated in Supplementary Fig. 17, the device exhibited values approaching the ideal benchmarks, with respective values of 49.31% for uniformity, 49.95% for inter-HD, and 0.99 for entropy.

For machine learning-based evaluation, we first generated a set of challenge–response pairs (CRPs) from a single PUF instance. The PUF response is implemented as a $250 \times 250$ binary matrix, in which each bit corresponds to a unique $(x, y)$ coordinate. Each challenge is formed by concatenating the 8-bit binary representation of the $x$-coordinate with that of the $y$-coordinate, resulting in a 16-bit challenge vector per location. The associated response bit is the value stored at the corresponding $(x, y)$ position in the matrix. This configuration inherently provides 62,500 unique CRPs per PUF instance. The 62,500 total bits were randomly divided into five subsets to generate independent CRP sets. Two standard classifiers—logistic regression (LR)[36] and support vector machines (SVM)[37]—were trained using different proportions of each CRP set (10 to 70% of the data) and tested on the remaining portions to evaluate their ability to infer unseen responses. The classification accuracy of the LR model is shown in Fig. 4f. Across all training conditions, the model failed to achieve prediction accuracy above random guessing[38], with performance consistently centered

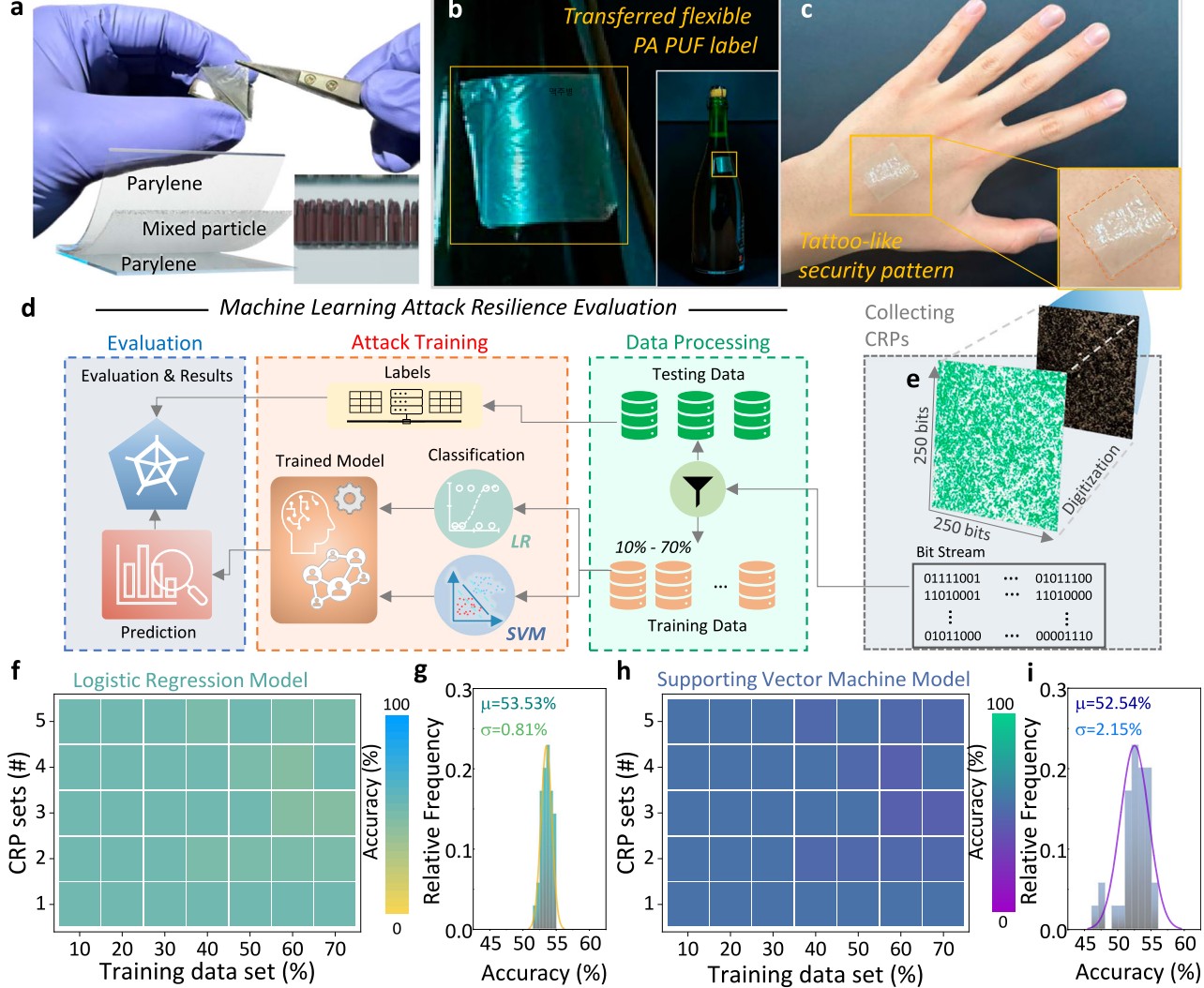

**Fig. 4 | Demonstration of conformable PA-PUF labels and evaluation of robustness against machine learning-based attacks. a** Photograph of the PA PUF label being peeled from a Si substrate, showing its flexibility and detachability. The corresponding layer structure is illustrated schematically. Photographs of PA PUF labels transferred to the surface of **b** a glass bottle and **c** human skin. **d** Flowchart describing the process used to evaluate resistance to machine learning-based attacks using supporting vector machine (SVM) and logistic regression (LR).

**e** Digitized 250 × 250 PA PUF pattern measured on human skin, demonstrating its suitability for conformable, on-skin electronics. **f** Prediction accuracy of LR across five challenge-response pair (CRP) sets with varying training data ratios ranging from 10 to 70%. **g** Distribution of prediction accuracy from LR method across multiple trials. **h** Prediction accuracy of SVM applied under the same conditions as in (**f**). **i** Statistical distribution of SVM prediction accuracy over multiple runs.

near 50%. The statistical distribution of LR accuracy values across multiple trials is illustrated in Fig. 4g, which follows a near-Gaussian profile with a mean of 53.53% and a standard deviation of 0.81%. A similar trend was observed with the SVM classifier. As shown in Fig. 4h, the accuracy remained nearly constant at ~50% regardless of training data volume, suggesting an absence of extractable structural features. This finding is supported by Fig. 4i, which shows the SVM accuracy distribution across multiple trials, again forming a Gaussian distribution curve centered at 52.54% with a standard deviation of 2.15%. Through the above characterization, we confirmed that the PA-PUF response patterns lack learnable structure and are effectively resistant to inference by conventional machine learning models. The centered and narrow distributions of prediction accuracy further support that PA-PUF signatures are statistically unpredictable and robust against model-based attacks. To further validate the modeling resistance of the PA-PUF under more advanced inference scenarios, we additionally conducted an attack simulation using a convolutional neural network (CNN) model (Supplementary Fig. 18). In this evaluation, each

response bit was treated as the ground truth label, and its corresponding challenge input was derived from the binary encoding of the ($x$, $y$) coordinates. The CNN architecture included convolutional and pooling layers with ReLU activation, followed by fully connected layers for binary classification. The model was trained in a supervised learning setting using varying training set sizes ranging from 10 to 70%, and five repeated trials were conducted to assess variability. As shown in Supplementary Fig. 18b–d, the CNN failed to exceed 53.3% prediction accuracy even at the highest training ratio, with an overall average accuracy of 52.39%. This result indicates that the proposed PA-PUF exhibits strong resistance not only against classical machine learning models, such as LR and SVM, but also against advanced neural network–based modeling attacks.

While resistance to modeling attacks is essential, real-world deployment further requires that the PA-PUF maintain stable performance under operational and environmental perturbations. To evaluate the environmental and operational robustness of the PA-PUF, we further conducted a comprehensive series of experiments simulating real-world

conditions, including repeated laser exposure, variation in laser parameters, ambient acoustic noise, and physiological factors such as skin temperature and in vivo motion. First, to assess the effect of prolonged laser exposure, a single location on the PA-PUF was irradiated with 20,000 consecutive pulses (8 ns pulse width, 20 kHz, 750 nJ). The resulting PA signal amplitude remained stable (367.25 ± 9.65 arbitrary units), indicating no degradation under repeated use (Supplementary Fig. 19). From a thermal standpoint, these laser parameters satisfy the condition for thermal confinement ($\tau\_pulse \ll \tau\_thermal$), and the inter-pulse interval (50 μs at 20 kHz) far exceeds the estimated heat diffusion time (~2.5 μs), ensuring that heat dissipates fully between pulses. To further examine pulse interval effects, we varied the PRF from 20 to 40 kHz while keeping energy per pulse constant. Across all PRF conditions, cryptographic performance remained unaffected, with inter-HD, entropy, and uniformity values near their respective ideals (49.18%, 0.9784, 49.94%), and similarity exceeding 95% (Supplementary Fig. 20). To assess robustness in wearable applications, the PUF was transferred to a flexible PET substrate and subjected to 100 bending cycles. Pre- and post-deformation responses showed strong agreement, with similarity exceeding 97% and negligible changes in randomness metrics (inter-HD ~49.03%, entropy ~0.9707, uniformity ~49.15%) (Supplementary Fig. 21). Thermal stability was also evaluated by heating the device to physiologically relevant temperatures (e.g., 36.5–50 °C). The resulting PA responses exhibited consistent entropy and uniformity values, and the bit-level similarity remained above 95% across conditions (Supplementary Fig. 22).

To further evaluate the influence of in vivo physiological motion—such as respiration, heartbeat, and minor posture shifts—on PUF stability, we have conducted repeated measurements while the device was attached to human skin as illustrated in Supplementary Fig. 23. Four PA images were acquired over time, with the first defined as the enrollment scan and the remaining three as subsequent scans. Despite natural subject movement, the resulting PUF responses exhibited consistent bit-level similarity, with an average bit error rate of 10.35%. Key statistical metrics (uniformity, entropy, and inter-HD) remained close to ideal values across all scans. To mitigate this level of variability and ensure robust key regeneration, we have incorporated a fuzzy extractor—a cryptographic primitive designed to generate stable keys from noisy physical inputs. During the enrollment phase, a key and associated helper data are derived from the initial PUF output. In subsequent authentication phases, the helper data enables recovery of the original key from noisy PUF responses, provided the bit error remains within a correctable range. This process ensures that the extracted key remains consistent even under in vivo fluctuations, without compromising security. Using this approach, we have successfully reconstructed the original cryptographic key from all subsequent scans with 100% accuracy. These results demonstrate the effectiveness of fuzzy extraction in compensating for motion-induced noise, reinforcing the practicality of the proposed PA-PUF system for secure wearable applications.

Finally, to evaluate ambient noise resilience, the system was tested under controlled background noise. Since the PA signals lie within the 5–30 MHz range and ambient noise sources typically fall below 20 kHz, spectral overlap is negligible. Furthermore, by applying a bandpass filter matched to the transducer bandwidth, we improved the signal-to-noise ratio from 19.06 to 25.46 dB (Supplementary Fig. 24), and the resulting cryptographic patterns remained highly consistent (similarity >95.04%) (Supplementary Fig. 25). These findings confirm that the proposed PA-PUF exhibits strong resilience to thermal, mechanical, acoustic, and physiological perturbations, highlighting its practical potential for secure and robust wearable applications.

## Discussion

In summary, we present an alternative concept of PUFs that convert light into sound to generate cryptographic keys. The proposed

PA-PUFs diverge from conventional optical or electronic PUF paradigms by introducing an inherently different physical transduction pathway from photons to phonons. This mechanism is enabled by a composite architecture combining CuO NPs as broadband photo-absorbers and $SnO_2$ NPs as a porous dispersive matrix, resulting in spatially random structures that emit unique acoustic signatures upon light stimulation. The binary responses extracted from photoacoustic maximum amplitude projection (PA-MAP) images satisfy key security metrics, including uniformity (49.54%), inter-device Hamming distance (49.69%), entropy (0.983), and bit aliasing (49.38%) across ten independently fabricated devices. These signatures exhibit platform-independent randomness and maintain cryptographic robustness even after mechanical transfer onto diverse substrates, including human skin. The PA-PUF's light-to-sound transduction pathway introduces machine-learning resistance, with both logistic regression and SVM models failing to infer bit values beyond random chance (≈50% accuracy). This confluence of material-driven unpredictability, physical flexibility, and signal-level security highlights the PA-PUF as a promising route towards secure, scalable, and wearable authentication platforms. We believe that this light-to-sound approach will inspire further investigation in physical cryptography and beyond.

## Methods

### Material preparation, nanoparticle synthesis, and device fabrication

Tin tetrachloride ($SnCl_4$, 98%), dichloromethane ($CH_2Cl_2$, ≥99.8%), tert-butanol ($C_4H_{10}O$, ≥99.5%), Tetramethylammonium hydroxide solution (TMAH, 1.0 M ± 0.02 M in $H_2O$), copper (II) acetate (98%), ethyl alcohol (≥99.9%), chloroform ($CHCl_3$, ≥99 %) were purchased from Sigma-Aldrich (USA). Metal oxide NPs were synthesized through a facile solvothermal approach. In the case of $SnO_2$ NPs, the precursor solution was prepared by dissolving 5.9 mL of $SnCl_4$ in 44.1 mL of dichloromethane under constant stirring for homogeneous mixing. The resulting solution was then transferred to a Teflon-lined autoclave and subjected to thermal treatment at 100 °C for 24 h in a convection oven to induce the formation of $SnO_2$ NPs via elimination 1 reaction between a metal halide and tertiary alcohol[39]. After cooling to room temperature, the obtained white precipitate was washed multiple times with acetone and ethanol to remove unreacted residues and side products. The final product was collected through centrifugation and dried under vacuum for subsequent use. In the case of the CuO NPs synthesis, 0.29 g of copper (II) acetate (a Cu precursor) was dissolved in 30 mL of ethanol under continuous stirring to ensure complete dissolution. In a separate container, 3.25 mL of TMAH was diluted in 6.75 mL of ethanol to prepare the base solution. The Cu precursor solution was heated to 75 °C under constant stirring at 600 rpm in an oil bath, and the TMAH solution was added dropwise at a rate of 1 mL·min⁻¹ over a total duration of 10 min to facilitate nucleation. The reaction was maintained for 2 h to allow complete conversion into CuO NPs. Following the synthesis, the CuO NPs were separated via centrifugation at 2683 × g and washed with ethanol to eliminate residual precursors and reaction byproducts. This purification step was repeated four times to ensure high purity. The collected NPs were then dried and stored for subsequent material characterization and device fabrication.

To prepare the NPs-based thin films, the synthesized CuO and $SnO_2$ NPs materials were dispersed in a chloroform:ethanol mixture (3:1 ratio) at a concentration of 60 mg·mL⁻¹. In the case of the mixed particle samples, each metal oxide NPs was re-dispersed in the same disperse medium with a final concentration of 60 mg·mL⁻¹. The prepared solutions were then deposited onto a 2.5 × 2.5 cm² glass substrate, which had been cleaned with isopropyl alcohol (IPA), ethanol, and deionized (DI) water. A volume of 416 μL of the solution was dispensed onto the substrate, followed by spin coating at 1500 RPM. The coated samples were then annealed on a hot plate at 120 °C for 10 min

to complete the fabrication process. To prevent physical delamination of the as-fabricated particles, parylene encapsulation layer was deposited on spin-casted NPs film by using parylene coater (OBT-PC300, Obang Technology Co., LTD, Gimpo, South Korea).

## Characterization of the synthesized NPs-based films

The morphological characteristics and elemental distribution of the thin films were characterized using a digital microscope (DM, AM4113T, AnMo Electronics, Taiwan), another bright-field microscope (BF, IX73, Olympus, Japan), X-ray photoelectron spectroscopy (XPS, ESCALAB250, Thermo Fisher Scientific Inc., Waltham, U.S.A) and a field-emission scanning electron microscope (FE-SEM, SU8600, Hitachi, Tokyo, Japan) equipped with an energy-dispersive spectrometer (EDS, SU8600, Hitachi, Tokyo, Japan). The size distribution and absorption profiles of the NPs were characterized with a dynamic light scattering measurement system (DLS, ELS-8000, Otsuka Electronics, Osaka, Japan) and UV-vis spectrophotometry (LAMDA 750, Perki-nElmer, Waltham, U.S.A). The surface characteristics and height profiles of the NPs films were investigated by using atomic force microscopy (AFM, Park NX10, Park Systems, Suwon, South Korea). To determine the optical bandgap energies, Tauc plots were constructed using the following relation:

$$(\alpha h\nu)^n = A\left(h\nu - E_g\right) \qquad (1)$$

where $\alpha$ is the absorption coefficient, h$\nu$ is the photon energy, $E_g$ is the optical bandgap, $A$ is a constant, and $n$ is the exponent depending on the nature of the transition (1/2 for direct allowed, 2 for indirect allowed transitions). The values of $E_g$ were extracted by extrapolating the linear portion of the $(\alpha h\nu)^n$ versus h$\nu$ plot to the x-axis intercept.

## Photoacoustic signal measurement and image reconstruction

We employed an optical-resolution photoacoustic microscope (OR-PAM, OptichoM, Opticho, South Korea) with a 532 nm nanosecond laser (DX-532-2, Photonics Industries International Inc., U.S.A) operating at 20 kHz with an 8 ns pulse width. The laser energy was adjusted using a half-wave plate polarizing beam splitter (VA5-532/M, Thorlabs, U.S.A), and aligned with two flat mirrors (PF10-03-P01, Thorlabs, U.S.A). The beam was then coupled into a multimode fiber (M64L01, Thorlabs, U.S.A) via a collimator (F240FC-532, Thorlabs, U.S.A) and focused by an objective lens (AC127-050-A, Thorlabs, U.S.A). The focused beam was reflected by a mirror (MRA10-G01, Thorlabs, U.S.A), passed through a RUT, and delivered to the target via a water-immersible 1-axis GM scanner (GVS001, Thorlabs, U.S.A). The sample was placed in a DI water tank using a petri dish. PA signals were collected with the RUT (25 MHz center frequency), amplified by two serial amplifiers (PHA-13LN+, Mini-Circuits, U.S.A) with a total gain of 48 dB, and digitized using a 12-bit high-speed digitizer (ATS9352, Alazar Technologies, Canada) at a 500 MS·s$^{-1}$. A DAQ board (PCIe-6321, National Instruments, U.S.A) synchronized the timing sequence and controlled the scanning system. To achieve a wide field of view for PA PUF imaging, we employed a hybrid scanning approach. Fast optical scanning along the x-axis was performed using the GM scanner, while slow translation along the y-axis was controlled by stepper motors (L-509, Physik Instrumente, Germany). The stepper motors operated along both the x and y axis, allowing precise sample positioning and enabling target adjustments independently of the GM scanner.

The PA signal acquisition follows a structured sequence to reconstruct the final maximum amplitude projection (MAP) images, which were used for all PA-PUF image analyses in this study. Each laser pulse excites a specific point within the sample, generating a depth-resolved A-line (A-scan), which represents the amplitude of the PA signal along the axial (z) direction. A sequence of A-lines acquired during a lateral scan by the GM scanner forms a cross-sectional image, known as a B-scan. By translating the sample along the y-axis using a motorized stage, multiple B-scans are collected to construct a 3D volumetric PA image. For image visualization, we extract a MAP from the 3D dataset. The MAP image is generated by selecting the highest PA signal intensity along the axial direction at each lateral position, providing a 2D representation of the PUF structure. Since MAP images effectively highlight the strongest absorptive features, they are widely used for structural analysis in PA imaging. This approach allows high-resolution PA-PUF imaging while maintaining efficient data acquisition and processing.

## Flexible PA PUF imaging on human in vivo

For in vivo PA PUF imaging, a reflection-mode OR-PAM system was used. All procedures were conducted following the regulations and guidelines approved by the Institutional Review Board (IRB) of Sung-kyunkwan University (approval no. 2025-03-061). We recruited a healthy male volunteer (26 years old) for PA PUF imaging of the hand. Informed consent was obtained after providing a detailed explanation of the procedure. The laser pulse energy used for PA PUF imaging was 750 nJ. The pulsed laser fluence of ~3.31 mJ·cm$^{-2}$ at 532 nm well below the ANSI safety limit for biological exposure[40].

## Randomness evaluation of the PUFs with figure of merits

The performance of the PUF responses was evaluated using the following standard statistical metrics[41–43]:

Uniformity ($U$) indicates the proportion of "1" bits in a single response vector $R_j = \{r_{j,1}, \ldots, r_{j,n}\}$, where $r_{j,i} \in \{0,1\}$.

$$U_j = \frac{1}{n}\sum_{i=1}^{n} r_{j,i} \times 100\% \qquad (2)$$

Inter-Hamming distance (inter-HD) is the average proportion of mismatched bits between all pairs of response vectors from $M$ different devices:

$$inter - HD = \frac{2}{M(M-1)}\sum_{j=1}^{M}\sum_{k=j+1}^{M}\left(\frac{1}{n}\sum_{i=1}^{n}\delta\left(r_{j,i}, r_{k,i}\right)\right) \times 100\% \qquad (3)$$

where $\delta(a,b) = 1$ if $a \neq b$, and 0 otherwise.

Entropy (Shannon Entropy, H) quantifies the randomness of each response:

$$H_j = -p_0\log_2 p_0 - p_1\log_2 p_1 \qquad (4)$$

where $p_0$ and $p_1$ are the empirical probabilities of "0" and "1" in $R_j$.

Bit-aliasing is defined as the average proportion of "1" values at bit position $i$ across all devices:

$$Bit - aliasing = \frac{1}{M}\sum_{j=1}^{M} r_{j,i} \times 100\% \qquad (5)$$

Intra-chip similarity was evaluated by repeated measurements on the same PUF device. The intra- Hamming distance (intra-HD) for $T$ trials is:

$$intra - HD^j = \frac{1}{T}\sum_{i=1}^{T}\left(\frac{1}{n}\sum_{i=1}^{n}\delta\left(r_{j,i}^{(1)}, r_{j,i}^{(t)}\right)\right) \times 100\% \qquad (6)$$

Inter-PUF correlation was computed using the Pearson correlation coefficient between pairs of response vectors:

$$\rho_{jk} = \frac{\sum_{i=1}^{n}\left(r_{j,i} - \bar{r}_k\right)\left(r_{k,i} - \bar{r}_k\right)}{\sqrt{\sum_{i=1}^{n}\left(r_{j,i} - \bar{r}_j\right)^2} \cdot \sqrt{\sum_{i=1}^{n}\left(r_{k,i} - \bar{r}_k\right)^2}} \qquad (7)$$

## Data availability

The authors declare that the data supporting the findings of this study are available within the article and its Supplementary Information file. Source data are provided with this paper.

## Code availability

The MATLAB (R2024b) codes employed for data processing is included with the source data in the Source data file.

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

## Acknowledgements

This work was supported by the National Research Foundation (NRF) grant (RS-2023-00210194 (H.Y.), RS-2024-00433166 (H.Y.), RS-2024-00442020 (H.Y.), RS-2023-00266110 (B.P.), RS-2024-00462912 (B.P.), RS-2023-00210682 (B.P.)) funded by the Ministry of Science and ICT (MSIT) of the Korean government and by the BK21 FOUR Project. This work was supported by Institute of Information & communications Technology Planning & Evaluation (IITP) under the artificial intelligence semiconductor support program to nurture the best talents (IITP-(2025)-RS-2023-00253914) grant funded by the Korea government (MSIT).

## Author contributions

H.Y. and B.P. initiated and supervised all the research. T.P., R.K., and J.K. conducted and designed the experimental work and data analysis. T.P. and R.K. synthesized the materials. The manuscript was written through the contributions of all authors. All authors have given approval to the final version of the manuscript.

## Competing interests

The authors declare no competing interests.
