## [Transparent Peer Review file · Nature Communications]

Light In, Sound Keys Out: Photoacoustic PUFs from Stochastic Nanocomposites

Corresponding Author: Professor Hocheon Yoo

Version 0:

Reviewer comments:

Reviewer #1

(Remarks to the Author)

Reviewer #2

(Remarks to the Author)

Title: Light In, Sound Keys Out: Photoacoustic PUFs from Stochastic Nanocomposites

In this paper, the authors propose a novel physical unclonable function (PUF) concept utilizing the photoacoustic (PA) effect from a stochastic nanocomposite of CuO and SnO₂ nanoparticles. The core idea involves generating unique acoustic signals upon laser excitation, which are then converted into cryptographic keys. This research presents material characterization and demonstrates the extraction of binary patterns, reporting performance metrics such as uniformity, inter-device Hamming distance, entropy, and resistance to basic machine learning attacks, along with mechanical transferability to various substrates. While the photoacoustic approach for PUFs introduces an interesting physical transduction mechanism, several key aspects of the material-to-response correlation, imaging limitations, data processing methodology, and robustness evaluation require substantial clarification and further assessment to fully validate the proposed system's performance and security claims. I expect this manuscript to become an outstanding paper after revisions are noted as below.

1. The authors adopt a 3:1 mass ratio of CuO to SnO₂ at 60 mg·mL⁻¹ for film deposition, yet no justification or optimization for these parameters is provided. Could the authors elaborate on how this specific ratio and concentration were selected and whether alternate ratios or concentrations were experimentally screened to maximize PA signal entropy and minimize aggregation effects? Including even a brief summary of any systematic optimization—either carried out here or cited from the literature—would more convincingly demonstrate that this formulation indeed maximizes both randomness generation and mechanical flexibility.
2. The OR-PAM lateral resolution (10 μm) far exceeds individual nanoparticle dimensions, implying that each pixel averages over many particles. Please discuss how this resolution limit and spatial averaging affect the PUF's unclonability, clarifying whether the observed randomness truly reflects nanoscale heterogeneity or is constrained by the imaging system's resolution. A concise examination of this point would help establish whether the PUF's security genuinely originates from material-level randomness rather than imaging artifacts.
3. The workflow bins 4×4 pixel blocks and applies a single global threshold to produce binary matrices, but metrics such as uniformity, entropy, and inter-HD can be highly sensitive to these parameters. Have the authors performed a sensitivity analysis (e.g., ±5 % threshold shifts or alternative block sizes) to confirm that the reported figures of merit remain stable? Presenting confidence intervals or variance data for small processing deviations would further reinforce the robustness and repeatability of the key-generation method.
4. The resistance evaluation is limited to logistic regression and SVM classifiers, yet advanced attackers often employ deep-

learning techniques. Have the authors considered testing convolutional neural networks (CNNs) or transformer-based models to infer PA-PUF patterns and exploring adversarial training to probe vulnerabilities? Including at least one pilot experiment with a more complex neural architecture—or a brief rationale for why simpler models suffice—would significantly strengthen the manuscript’s security claims.

5. Transfer onto human skin is demonstrated, but in-vivo conditions (skin temperature fluctuations, hydration changes, and movement) could introduce bit errors. How might these physiological factors influence the bit-error rate and overall reliability in a wearable application? Reflecting on these real-world influences and proposing mitigation strategies—such as adaptive thresholding or real-time calibration—would underscore the feasibility of practical wearable use.

6. Ambient acoustic noise in non-laboratory settings (e.g., machinery, human activity) could corrupt PA signals and key extraction. Have the authors characterized the system’s signal-to-noise ratio under controlled noise sources and implemented filtering or time-gating techniques to reject background interference? Demonstrating noise-immunity thresholds through such tests would convincingly illustrate the approach’s practicality outside idealized environments.

7. The PA response intensity strongly depends on local film thickness and nanoparticle dispersion, so uneven coatings could bias bit distributions. Can the authors provide profilometry, ellipsometry, or cross-sectional SEM data to verify thickness uniformity across the 2.5×2.5 cm² substrate and quantify acceptable non-uniformity tolerances? Outlining coating-optimization strategies (e.g., optimized spin- versus flow-coating parameters) would substantiate the method’s scalability and ensure consistent key generation over large areas.

Version 1:

Reviewer comments:

Reviewer #1

(Remarks to the Author)

N/A

Reviewer #2

(Remarks to the Author)

I would like to express my sincere appreciation to the authors for their exceptionally meticulous and comprehensive responses to the reviewers’ questions.

In particular, they have offered scientifically rigorous reflections on each point and resolved numerous concerns by validating their explanations with additional experimental evidence.

Moreover, by incorporating appropriate clarifications into the main text and providing supplementary supporting materials, they have clearly articulated and strongly reinforced their arguments.

In light of these thorough revisions, I believe the current version of the manuscript is fully suitable for publication.

NCOMMS-25-25989

Light In, Sound Out: Photoacoustic PUFs from Stochastic Nanocomposites

This manuscript is well described with abundant figures and explanations about a physical unclonable function based on the photoacoustic (PA) effect of the mixed nanoparticles and its unique properties. The suggested PUF is reasonable presenting about its randomness with sufficient demonstration of quality factors and entropy. Furthermore, the ability of defending machine learning attacks can effectively suggest excellent secure performance. However, there are several questions that the author should consider before publication.

1. Sufficient explanation is required to verify sufficient CRP space size made from the PUF. For instance, to identify all individuals globally, the space must exceed the world population of approximately 8 billion. Considering multiple devices, the required number increases even further. How can the CRP space of the proposed PUF be quantitatively calculated?

2. Additional demonstration is required as to whether the given materials exhibit a direct bandgap characteristic or an indirect bandgap characteristic. When calculating the $(\alpha h\nu)^n$ value, information about not only the direct bandgap ($n=2$), but also that about the indirect bandgap ($n=1/2$) can be obtained. By analyzing the UV-Vis absorption data of the two materials for indirect transitions, and by examining how phonon interactions contribute to light-matter interactions, the optical properties can be more thoroughly characterized.

3. A discussion on the stability of the proposed PUF should be added. How does the PUF behave under repeated laser exposure at the same position? Would prolonged exposure to laser pulses lead to a change in the PUF's characteristics? If changes occur, what is the required cooling time between laser pulses to maintain stable responses? Additionally, how does the laser pulse interval affect the reproducibility of the PUF characteristics? Beyond this, could the physical label made from this PUF be vulnerable to mechanical abrasion or temperature changes? Further experimental investigations into these aspects would greatly enhance the work.

4. The authentication time of the PUF should be addressed. In general, shorter authentication times (preferably less than milliseconds) are advantageous for PUF applications. Optical PUFs, in particular, often exhibit rapid responses. Including data on the authentication time would help substantiate the advantages of the proposed PUF.

5. The following reference paper would support:

1) G. G. Yang, H. J. Choi et al., *Nature Reviews Electrical Engineering* **2024**, 1(2): 124-138.

2) G. S. Lee et al., *Advanced Materials* **2024**, 36(11): 2307689

Manuscript ID: NCOMMS-25-25989

Title: Light In, Sound Keys Out: Photoacoustic PUFs from Stochastic Nanocomposites

Dear Reviewers,

We, the authors, sincerely appreciate the time and effort you have dedicated to reviewing our manuscript. We have carefully considered all the questions and comments provided by the reviewers and have prepared a detailed point-by-point response letter, along with a thoroughly revised manuscript.

We have revised the manuscript in accordance with the Reviewers' valuable feedback and hope that the changes will meet your expectations for further consideration of our work as a research article in Nature Communications.

For your convenience, the Reviewers' comments are presented in **black**, while our responses are provided in blue. All revisions made to the manuscript are clearly **highlighted**.

Thank you once again for your constructive feedback and for your continued support in the evaluation of our manuscript.

Sincerely,

Hocheon Yoo

Reviewer (#1)'s COMMENTS:

This manuscript is well described with abundant figures and explanations about a physical unclonable function based on the photoacoustic (PA) effect of the mixed nanoparticles and its unique properties. The suggested PUF is reasonable presenting about its randomness with sufficient demonstration of quality factors and entropy. Furthermore, the ability of defending machine learning attacks can effectively suggest excellent secure performance. However, there are several questions that the author should consider before publication.

Response:

We sincerely thank the Reviewer #1 for thoroughly reading our manuscript and providing constructive and valuable comments. We have carefully revised the manuscript in response and prepared detailed point-by-point responses to each comment, as outlined below.

R1Q1. Sufficient explanation is required to verify sufficient CRP space size made from the PUF. For instance, to identify all individuals globally, the space must exceed the world population of approximately 8 billion. Considering multiple devices, the required number increases even further. How can the CRP space of the proposed PUF be quantitatively calculated?

Response:

We appreciate the Reviewer #1's valuable and constructive feedback regarding the challenge-response pair (CRP) space and the scalability of the proposed photoacoustic physical unclonable function (PA-PUF) device. In our work, each CRP is composed by associating the binary (8-bit) representations of the (x, y) coordinates of a 250×250 bit matrix (resulting in a 16-bit challenge) with the corresponding response bit at that location. As a result, a single PUF device inherently generates 62,500 unique CRPs. To ensure the CRP space exceeds the global population of approximately 8 billion individuals, at least 128,000 devices would be required ($8 \times 10^9 \div 62,500 = 128,000$). In practice, this number is readily scalable by fabricating device arrays on wafer-scale substrates, or by increasing the resolution of the addressable matrix (e.g., 512×512 or 1024×1024), which would proportionally expand the CRP space without increasing device footprint.

We thank the Reviewer #1 for pointing out this important aspect, which was not sufficiently addressed in the original manuscript. We have now included a clarification of the CRP structure and its scale in the revised manuscript as follows:

Revised Manuscript (PAGE 13)

→ [R1Q1] For machine learning-based evaluation, we first generated a set of challenge-response pairs (CRPs) from a single PUF instance. The PUF response is implemented as a 250×250 binary matrix, in which each bit corresponds to a unique (x, y) coordinate. Each challenge is formed by concatenating the 8-bit binary representation of the x-coordinate with that of the y-coordinate, resulting in a 16-bit challenge vector per location. The associated response bit is the value stored at the corresponding (x, y) position in the matrix. This configuration inherently provides 62,500 unique CRPs per PUF instance.

R1Q2. Additional demonstration is required as to whether the given materials exhibit a direct bandgap characteristic or an indirect bandgap characteristic. When calculating the $(\alpha h\nu)^n$ value, information about not only the direct bandgap ($n=2$), but also that about the indirect bandgap ($n=1/2$) can be obtained. By analyzing the UV-Vis absorption data of the two materials for indirect transitions, and by examining how phonon interactions contribute to light-matter interactions, the optical properties can be more thoroughly characterized.

Response:

We sincerely thank the Reviewer #1 for the constructive suggestion. As the Reviewer #1 commented, we have further investigated the indirect bandgap characteristics of the materials by conducting additional analyses based on ultraviolet-visible (UV-Vis) absorption measurements. During the process of extracting the indirect bandgap from the original data, we identified certain discontinuities in the UV-Vis spectra, which we attribute to instrumental noise and measurement fluctuations (Figure R1a). To ensure the reliability and clarity of the bandgap identification, we have conducted a new set of UV-Vis absorption measurements under optimized conditions (Figure R1b). The newly acquired spectra were obtained without observable noise artifacts, and the extracted bandgap values and trends are consistent with those reported in our original results.

As shown in Figure R2, the extracted optical properties indicate that CuO nanoparticles (NPs) possess an indirect bandgap of 1.24 eV and a direct bandgap of 1.64 eV, while SnO₂ exhibits a larger indirect bandgap of 3.21 eV and a direct bandgap of 4.29 eV. Our additional UV-Vis analyses confirm that both CuO and SnO₂ display signatures of both direct and indirect transitions. In our system, the excitation wavelength used for PA signal generation is 532 nm (2.33 eV), which exceeds both the direct and indirect bandgaps of CuO. Thus, photoexcitation in CuO can proceed via both direct transitions and phonon-assisted indirect processes. The presence of an indirect bandgap is particularly advantageous for PA signal generation, as phonon-assisted non-radiative transitions promote the conversion of absorbed photon energy into localized heat. In materials with indirect bandgaps, electronic transitions require momentum conservation via phonon participation, increasing the likelihood of non-radiative recombination [R1]. This non-radiative process leads to lattice heating rather than photon emission, and the resulting thermoelastic expansion serves as the physical origin of the photoacoustic effect [R2, R3].

In contrast, for SnO₂ NPs, the excitation energy is insufficient to induce either direct or indirect electronic transitions, and thus SnO₂ primarily serves to introduce spatial randomness without contributing to the PA signal. This additional analysis further confirms our original design: CuO NPs are responsible for PA signal generation via direct optical transitions, while SnO₂ NPs function as a non-active matrix providing spatial diversity.

Once again, we deeply appreciate the Reviewer #1's valuable and constructive comments. We have incorporated this analysis and the corresponding discussion into the revised manuscript as follows:

Revised Manuscript (PAGE 8)

→ [R1Q2] Tauc plot analysis (Supplementary Fig. 6) revealed both direct and indirect optical transitions in CuO (1.64 eV and 1.24 eV, respectively), while SnO₂ exhibited a large direct bandgap of 4.29 eV and an indirect transition at 3.21 eV. The photon energy used for PA excitation (532 nm, 2.33 eV) exceeds both bandgaps of CuO, allowing absorption through both direct and phonon-assisted indirect transitions. In contrast, the excitation energy is insufficient for electronic transitions in SnO₂, indicating that SnO₂ serves primarily as a spatial randomizing matrix rather than a PA-active component.

Revised Manuscript (PAGE 17-18)

→ [R1Q2] To determine the optical bandgap energies, Tauc plots were constructed using the following relation:

$$(\alpha hv)^n = A(hv - E_g)$$

where α is the absorption coefficient, hv is the photon energy, E_g is the optical bandgap, A is a constant, and n is the exponent depending on the nature of the transition (1/2 for direct allowed, 2 for indirect allowed transitions). The values of E_g were extracted by extrapolating the linear portion of the $(\alpha hv)^n$ versus hv plot to the x-axis intercept.

[R1] Zelewski, S. J., & Kudrawiec, R. (2017). *Scientific Reports*, 7(1), 15365.

[R2] Palzer, S. (2020). *Sensors*, 20(9), 2745.

[R3] Misztal, K., Kopaczek, J., & Kudrawiec, R. (2025). *Photoacoustics*, 41, 100668.

Figure R1. Absorption spectra of the CuO, SnO₂, and mixed NP films: (a) represented in the original manuscript and (b) newly measured results.

Figure R2 (Figure S4). Tauc plot analysis of CuO and SnO₂ NPs films to determine their optical bandgap energies. (a) Plot showing the estimated indirect bandgap of CuO NPs (1.24 eV). (b) Indirect bandgap of SnO₂ NPs determined to be 3.21 eV. (c) Direct bandgap of CuO NPs extracted from the linear region of the Tauc plot, yielding 1.64 eV. (d) Direct bandgap of SnO₂ NPs estimated as 4.29 eV.

R1Q3. A discussion on the stability of the proposed PUF should be added. How does the PUF behave under repeated laser exposure at the same position? Would prolonged exposure to laser pulses lead to a change in the PUF's characteristics? If changes occur, what is the required cooling time between laser pulses to maintain stable responses? Additionally, how does the laser pulse interval affect the reproducibility of the PUF characteristics? Beyond this, could the physical label made from this PUF be vulnerable to mechanical abrasion or temperature changes? Further experimental investigations into these aspects would greatly enhance the work.

Response:

We appreciate the Reviewer #1's thoughtful and multifaceted comments regarding the stability and robustness of the proposed PUF under repeated laser exposure. The Reviewer #1's suggestions, including the investigation of potential thermal accumulation, the effect of laser pulse intervals, long-term structural stability, and vulnerability to environmental or mechanical disturbances, provided valuable guidance for improving the clarity and completeness of our study. We have carefully addressed each aspect through a combination of theoretical justification (e.g., thermal confinement conditions), quantitative analysis (e.g., intra-HD stability metrics), and additional experimental data. Our response is organized as follows:

(i) Repeated laser exposure at the same position:

We used a laser with an 8 ns pulse width, 20 kHz repetition rate, and 750 nJ pulse energy. To evaluate the signal stability under repeated exposure, we irradiated the same location 20,000 times and measured the resulting PA signal amplitude. The average signal amplitude was 367.25 ± 9.65 , indicating a highly stable response (Figure R3). Additionally, as described on page 9 of the original manuscript, we acquired and analyzed 100 repeated PA images at the same region of interest (ROI). The resulting intra-HD indicated an average bit-level agreement of 9% (Figure 3j of the original manuscript), reflecting considerable repeatability and robustness.

(ii) If changes occur, what is the required cooling time between laser pulses to maintain stable responses?:

This can be addressed in the context of thermal confinement. Thermal confinement is satisfied when the laser pulse duration is shorter than the thermal diffusion time of the

absorber (e.g., nanoparticle film), defined as:

$$\tau_{thermal} = d^2 / 4\alpha$$

Where d is the heat diffusion length ($\sim 1 \mu\text{m}$ for nanoscale materials), and α is the thermal diffusivity ($\sim 1 \times 10^{-7} \text{ m}^2/\text{s}$ for many solids). Assuming $d = 1 \mu\text{m}$,

$$\tau_{thermal} = 2.5 \mu\text{s}$$

Since our pulse width is $8 \text{ ns} \ll 2.5 \mu\text{s}$, we satisfy thermal confinement — meaning heat does not accumulate during each pulse and has time to dissipate before the next. Additionally, the interval between pulses with 20 kHz is:

$$T_{rep} = 1 / 20000 = 50 \mu\text{s}$$

This interval is 20 times longer than the thermal diffusion time, giving ample time for heat to dissipate between pulses and preventing thermal buildup that could lead to damage.

(iii) How does the laser pulse interval affect the reproducibility of the PUF characteristics?:

PA signal generation follows the fundamental equation:

$$p_0 = \Gamma \mu_a F$$

Where p_0 is the initial pressure, Γ is the Gruneisen parameter, μ_a is the optical absorption coefficient, and F is the laser fluence. Among these parameters, the only laser-dependent term is F , which is directly proportional to the pulse energy. Therefore, when all other parameters are held constant, variations in pulse energy will result in proportional changes in the photoacoustic signal amplitude. In contrast, the pulse interval (or pulse repetition frequency, PRF) does not affect the initial pressure as long as the pulse energy remains constant. This implies that changing the laser repetition rate does not influence the PUF characteristics, provided that the pulse energy per shot is stable. To validate this, we have newly conducted additional experiments to analyze the stability of PUF characteristics under PRF of 20, 30, and 40 kHz. For each case, we calculated key metrics including inter-HD, entropy, uniformity, and similarity. As represented in Figure R4, the inter-HD, entropy, and uniformity across different PRF

conditions remained high at approximately 49.18%, 0.9784, and 49.94%, respectively. The similarity remained above 95% for all PA responses. This is further supported by our experimental results, which show consistent PUF responses under varying PRFs but fixed pulse energy conditions.

(iv) Could the physical label made from this PUF be vulnerable to mechanical abrasion or temperature changes?

We have conducted a series of experiments to evaluate the robustness and stability of the PUF's physical structure and associated cryptographic features under these conditions. To assess mechanical durability, the PUF device was affixed to a flexible PET substrate and subjected to 100 repetitive bending cycles. PA signals were acquired both before and after the mechanical deformation (Figure R5). For each condition, we computed key statistical metrics (including inter-HD, entropy, uniformity, and similarity) to evaluate the consistency of the PUF responses. The results showed that the inter-HD, entropy, and uniformity remained stable at approximately 49.03%, 0.9707, and 49.15%, respectively. The similarity between pre- and post-deformation responses exceeded 97%, indicating negligible impact on the PUF characteristics due to mechanical abrasion.

To investigate thermal robustness, we exposed the PUF device to a range of temperatures using a heating pad integrated into the sample stage (Figure R6). The PA responses were again analyzed using the same statistical criteria. The similarity across different temperature conditions consistently remained above 95%, confirming that temperature variations did not significantly alter the PUF outputs. These results collectively demonstrate that the proposed PUF device exhibits considerable resilience to both mechanical and thermal stress.

We sincerely thank Reviewer #1 for this insightful comment, which helped us improve the clarity and completeness of the manuscript. We have included a clarification of the robustness of the PA-PUF system in the revised manuscript as follows.

→ [R1Q3] While resistance to modeling attacks is essential, real-world deployment further requires that the PA-PUF maintain stable performance under operational and environmental perturbations. To evaluate the environmental and operational robustness of the PA-PUF, we further conducted a comprehensive series of experiments simulating real-world conditions, including repeated laser exposure, variation in laser parameters, ambient acoustic noise, and physiological factors such as skin temperature and in-vivo motion. First, to assess the effect of prolonged laser exposure, a single location on the PA-PUF was irradiated with 20,000 consecutive pulses (8 ns pulse width, 20 kHz, 750 nJ). The resulting PA signal amplitude remained stable (367.25 ± 9.65 arbitrary units), indicating no degradation under repeated use (Supplementary Fig. 19). From a thermal standpoint, these laser parameters satisfy the condition for thermal confinement ($\tau_{\text{pulse}} \ll \tau_{\text{thermal}}$), and the inter-pulse interval (50 μs at 20 kHz) far exceeds the estimated heat diffusion time ($\sim 2.5 \mu\text{s}$), ensuring that heat dissipates fully between pulses. To further examine pulse interval effects, we varied the PRF from 20 to 40 kHz while keeping energy per pulse constant. Across all PRF conditions, cryptographic performance remained unaffected, with inter-HD, entropy, and uniformity values near their respective ideals (49.18%, 0.9784, 49.94%), and similarity exceeding 95% (Supplementary Fig. 20). To assess robustness in wearable applications, the PUF was transferred to a flexible PET substrate and subjected to 100 bending cycles. Pre- and post-deformation responses showed strong agreement, with similarity exceeding 97% and negligible changes in randomness metrics (inter-HD $\sim 49.03\%$, entropy ~ 0.9707 , uniformity $\sim 49.15\%$) (Supplementary Fig. 21). Thermal stability was also evaluated by heating the device to physiologically relevant temperatures (e.g., $36.5^\circ\text{C} - 50^\circ\text{C}$). The resulting PA responses exhibited consistent entropy and uniformity values, and the bit-level similarity remained above 95% across conditions (Supplementary Fig. 22).

Figure R3 (Figure S19). Stability of PA signal amplitude under repeated laser exposure. Photoacoustic signal stability at a fixed location is over 20,000 repeated laser irradiations.

Figure R4 (Figure S20). Effect of laser frequency on randomness characteristics. (a) Schematic of the experimental setup for evaluating the stability of PA responses under different laser pulse intervals. (b) Digitized PA PUF patterns extracted from individual PA images. (c) Similarity analysis of PA responses obtained from repeated measurements at the same region of a single PUF device. (d) Statistical analysis of PA PUF responses, including average uniformity, entropy, and inter-HD.

Figure R5 (Figure S21). Mechanical robustness of the PA-PUF device. (a) Photographs of the experimental setup used for the mechanical stability test. (b) Digitized PA PUF patterns extracted from individual PA images. (c) Mean inter-HD distribution of PA PUF responses. (d) Entropy analysis results of PA PUF responses. (e) Uniformity distribution of PA PUF responses. A single PUF device was repeatedly tested at the same region before and after mechanical deformation.

Figure R6 (Figure S22). Thermal stability evaluation of the PA-PUF response. (a) Schematic illustration of the experimental setup for thermal stability testing. (b) Digitized PA PUF patterns extracted from each PA image. (c) Similarity analysis of each PA PUF response. A single PUF device was relatedly tested at the same region. (d) Uniformity, entropy, and inter-HD analysis results of each PA PUF responses.

R1Q4. The authentication time of the PUF should be addressed. In general, shorter authentication times (preferably less than milliseconds) are advantageous for PUF applications. Optical PUFs, in particular, often exhibit rapid responses. Including data on the authentication time would help substantiate the advantages of the proposed PUF.

Response:

We appreciate the Reviewer #1's valuable suggestion regarding the authentication time of the proposed PUF system. In our case, PA PUF exhibits a pixel generation speed of approximately 10^4 pixels per second over a $3 \times 3 \text{ mm}^2$ field of view, which corresponds to a high pixel generation density comparable to optical PUFs.

We thank Reviewer #1 for the helpful suggestion. We have added a clarification of the per-unit-area pixel generation speed to better highlight the system's rapid response characteristics.

Revised Manuscript (PAGE 4)

→ [R1Q4] During acquisition, each response pattern was obtained at a rate of approximately 10^4 pixels per second across a $3 \times 3 \text{ mm}^2$ area.

R1Q5. The following reference paper would support: 1) G. G. Yang, H. J. Choi et al., Nature Reviews Electrical Engineering 2024, 1(2): 124-138. 2) G. S. Lee et al., Advanced Materials 2024, 36(11): 2307689

Response:

We greatly appreciate the Reviewer #1's constructive and valuable comments. As suggested by the Reviewer #1, we have carefully reviewed the recommended and related references that support our work from the perspectives of flexible hardware platforms beyond the limitations of rigid silicon-based electronics, as well as material-level randomness.

We thank Reviewer #1 for the valuable suggestion and for providing the relevant references. We have incorporated a new discussion into the revised manuscript based on these insights.

Revised Manuscript (PAGE 3)

→ **[R1Q5]** Their reliance on rigid wafer-based substrates also imposed limitations, restricting applications in wearable⁶, flexible⁷, and embedded security systems⁸ that demand adaptability beyond conventional silicon platforms. Recent shifts toward adaptive, physically interactive hardware platforms further highlight this limitation⁹.

Revised Manuscript (PAGE 3)

→ **[R1Q5]** Similarly, block copolymer self-assembly has been shown to produce reproducible yet spatially diverse nanoscale patterns, offering a scalable route to encode physical randomness at the material level^{7,15}

Reviewer (#2)'s COMMENTS:

In this paper, the authors propose a novel physical unclonable function (PUF) concept utilizing the photoacoustic (PA) effect from a stochastic nanocomposite of CuO and SnO₂ nanoparticles. The core idea involves generating unique acoustic signals upon laser excitation, which are then converted into cryptographic keys. This research presents material characterization and demonstrates the extraction of binary patterns, reporting performance metrics such as uniformity, inter-device Hamming distance, entropy, and resistance to basic machine learning attacks, along with mechanical transferability to various substrates. While the photoacoustic approach for PUFs introduces an interesting physical transduction mechanism, several key aspects of the material-to-response correlation, imaging limitations, data processing methodology, and robustness evaluation require substantial clarification and further assessment to fully validate the proposed system's performance and security claims. I expect this manuscript to become an outstanding paper after revisions are noted as below.

Response:

We appreciate the Reviewer #2 for the thorough review and the positive, constructive comments. We have carefully revised the manuscript and prepared point-by-point responses to each comment, as outlined below.

R2Q1. The authors adopt a 3:1 mass ratio of CuO to SnO₂ at 60 mg·mL⁻¹ for film deposition, yet no justification or optimization for these parameters is provided. Could the authors elaborate on how this specific ratio and concentration were selected and whether alternate ratios or concentrations were experimentally screened to maximize PA signal entropy and minimize aggregation effects? Including even a brief summary of any systematic optimization—either carried out here or cited from the literature—would more convincingly demonstrate that this formulation indeed maximizes both randomness generation and mechanical flexibility.

Response:

We appreciate the Reviewer #2 for providing valuable and constructive suggestions. First, we would like to clarify a possible misunderstanding that may have led to this question. The 3:1 mass ratio mentioned in the original manuscript refers not to the CuO:SnO₂ nanoparticle (NP) ratio, but to the volume ratio of the mixed solvents (chloroform and ethanol) used for NP dispersion. As for the CuO and SnO₂ NP ratio in the coating solution formulation, we used a 1:1 mass ratio.

In response to the Reviewer #2's constructive suggestion, during this revision, we have additionally performed a series of experiments to systematically investigate the effect of nanoparticle mixing ratio and concentration on the randomness and aggregation behavior of the PA-PUF device. The CuO and SnO₂ NP coating solution were prepared at 60 mg·mL⁻¹ and combined in three different CuO:SnO₂ mass ratios (*i.e.*, 2:1, 1:1, and 1:2). As shown in Figure R7a–c, increasing the proportion of SnO₂ (1:2 condition) reduced the amount of PA-active CuO NPs, leading to a lower proportion of '1' values in the binary pattern. Conversely, when the CuO content was dominant (2:1), the excessive PA signal generation caused the binary output to be overly biased toward '1'. Only in the 1:1 condition did we observe a statistically balanced distribution of '0' and '1'.

This observation was further supported by the uniformity analysis shown in Figure R7d–f. The average uniformity values under the three CuO:SnO₂ ratios (1:2, 1:1, 2:1) were 4.33%, 48.46%, and 95.12%, respectively, confirming that only the 1:1 condition approached the statistically ideal value of 50%. Furthermore, this balanced formulation also achieved near-ideal results in other metrics such as an average inter-Hamming distance (inter-HD) of 49.4% and entropy of 0.99 (Figure R7g-I, and R7j-l). The above experimental results confirms that

our selected formulation simultaneously maximizes entropy and minimizes excessive aggregation effects.

In addition to compositional ratio, we have also evaluated the effect of total NP concentration on randomness characteristics while maintaining a fixed 1:1 CuO:SnO₂ mass ratio (Figure R8). Since the relative ratio of PA-active (CuO) and PA-inactive (SnO₂) NPs remains constant, the spatial distribution patterns of the binary maps are structurally similar across concentrations; however, the absolute PA signal intensity—which scales directly with the absolute CuO content—becomes the dominant differentiating factor. At low concentration (20 mg·mL⁻¹), insufficient PA signal yields a pronounced bias toward ‘0’ bits, giving an average uniformity of 35.2%, an inter-HD of 44.0%, and entropy of 0.931. At the high concentration (180 mg·mL⁻¹) extreme, signal saturation biases the output toward ‘1’ bits, resulting in uniformity of 90.8 %, Inter-HD of 16.6%, and entropy of 0.432. Only the intermediate concentration (60 mg·mL⁻¹) achieves a well-balanced PA output (uniformity 47.8%, inter-HD 49.7%, and entropy 0.987) all metrics approaching their statistical ideals. These results confirm that, under a fixed 1:1 NP ratio, the PA signal intensity governed by total concentration is the key factor controlling randomness quality, and that 60 mg·mL⁻¹ provides the optimal balance between under- and over-saturation.

Once again, we are grateful to Reviewer #2 for the valuable comments, which helped improve the depth and clarity of our work. We have added the above discussion and experimental results to our revised manuscript as follows.

Revised Manuscript (PAGE 11)

→ [R2Q1] The above random metrics were achieved by optimizing both the CuO:SnO₂ mass ratio and the total NP concentration, based on the interplay between PA signal strength and spatial dispersion. As represented in Supplementary Fig. 13, at a fixed total concentration of 60 mg·mL⁻¹, films were prepared with mass ratios of 2:1, 1:1, and 1:2. Increasing the CuO content enhanced signal intensity but also led to pixel saturation and an overrepresentation of ‘1’ bits in the binarized output. In contrast, increasing the SnO₂ fraction diluted the PA-active component, resulting in weaker signals and a bias toward ‘0’. Only the 1:1 ratio maintained a

sufficient PA response while ensuring uniform spatial randomness, yielding near-ideal values in uniformity, inter-HD, and entropy. A separate concentration sweep (20–180 mg·mL⁻¹) performed at the 1:1 ratio further confirmed that signal intensity must remain within an optimal range to preserve response quality. At low concentrations, the reduced number of absorbers lowered PA amplitude, leading to sparsely populated bitstreams. At high concentrations, signal saturation introduced strong local contrast and clustering artifacts. The intermediate value of 60 mg·mL⁻¹ achieved the best balance, producing stable bit distributions with minimal aggregation-induced bias. These observations confirm that both PA signal generation and randomness characteristics are strongly governed by material formulation (Supplementary Fig. 14).

Figure R7 (Figure S13). Effect of mixed nanoparticle solution ratio (CuO:SnO₂) on the randomness characteristics of PA-PUFs. (a–c) Optical and binarized PA-PUF response maps obtained from films fabricated by mixing CuO and SnO₂ NP coating solutions at volume ratios of 1:2, 1:1, and 2:1, respectively. Both CuO and SnO₂ solutions were prepared at the same particle concentration of 60 mg ml⁻¹ prior to mixing. (d–f) Distribution of uniformity (%) for each mixing condition, compared with the statistical ideal of 50%. (g–i) Inter-HD distributions, reflecting the degree of response variability across devices. (j–l) Device-wise entropy measurements for each composition, showing the highest entropy (0.990) at a 1:1 mixing ratio, while entropy is suppressed at 1:2 and 2:1.

Figure R8 (Figure S14). Effect of NP coating solution concentration on spatial randomness and uniformity of PA-PUFs. (a–c) Binarized PA-PUF response patterns derived from films fabricated using NP coating solutions at concentrations of 20, 60, and 180 mg·ml⁻¹, respectively. (d–f) Distribution of response uniformity across devices for each concentration condition, compared against the statistical ideal of 50%. (g–i) Inter-HD distributions reflect device-to-device variability in response patterns derived from coating solution concentration variation. (j–l) Entropy values measured for individual devices, illustrating bit-level unpredictability.

R2Q2. The OR-PAM lateral resolution (10 μm) far exceeds individual nanoparticle dimensions, implying that each pixel averages over many particles. Please discuss how this resolution limit and spatial averaging affect the PUF's unclonability, clarifying whether the observed randomness truly reflects nanoscale heterogeneity or is constrained by the imaging system's resolution. A concise examination of this point would help establish whether the PUF's security genuinely originates from material-level randomness rather than imaging artifacts.

Response:

We are very grateful for the Reviewer #2's insightful comments. We acknowledge that the lateral resolution of our OR-PAM system ($\sim 10 \mu\text{m}$) is significantly larger than the size of individual CuO and SnO₂ NPs ($< 100 \text{ nm}$), resulting in spatial averaging within each pixel. However, our objective is not to resolve individual NPs, but rather to extract unique optical-acoustic responses arising from stochastic mesoscale aggregation patterns as represented in Figure R9. These patterns, formed through inherently random processes during film deposition, are highly sensitive to initial conditions and effectively irreproducible, even under nominally identical fabrication protocols. The OR-PAM resolution is sufficient to capture these mesoscale features, which serve as the physical source of entropy for the PUF. Thus, we believe that the spatial averaging does not undermine, but instead appropriately reflects, the intended randomness and unclonability of our system. We also agree that future studies could benefit from improving the optical resolution toward the nanoscale, to more directly probe the underlying particle-level heterogeneity. Integrating advanced imaging modalities (such as super-resolution PA techniques or hybrid systems) may enable more detailed analysis of nanostructural features. We consider this an important direction for expanding the understanding and applicability of PUFs based on nanoparticle assemblies.

We thank the Reviewer for the thoughtful comment. We have added a brief note along with a schematic illustration in the revised manuscript to clarify that the observed randomness originates from mesoscale aggregation rather than individual nanoparticle resolution, as follows.

→ [R2Q2] While the spatial resolution of the OR-PAM system ($\sim 10\ \mu\text{m}$) exceeds the size of individual CuO and SnO₂ nanoparticles ($< 100\ \text{nm}$), the primary source of randomness is not individual particle contrast, but the emergent mesoscale aggregation patterns formed during film deposition (Supplementary Fig. 7). These stochastic structures are highly sensitive to fabrication conditions and effectively irreproducible, providing a robust entropy source for the PUF.

Figure R9 (Figure S7). Schematic illustration of the laser excitation geometry and spatial relationship between the beam size and nanoparticle distribution in the PUF device. The pulsed laser irradiates a heterogeneous surface composed of randomly aggregated CuO and SnO₂ nanoparticles. The beam waist ($\sim 10\ \mu\text{m}$) far exceeds the size of individual nanoparticles ($< 100\ \text{nm}$), resulting in spatial averaging within each pixel of the OR-PAM system.

R2Q3. The workflow bins 4×4 pixel blocks and applies a single global threshold to produce binary matrices, but metrics such as uniformity, entropy, and inter-HD can be highly sensitive to these parameters. Have the authors performed a sensitivity analysis (e.g., ±5 % threshold shifts or alternative block sizes) to confirm that the reported figures of merit remain stable? Presenting confidence intervals or variance data for small processing deviations would further reinforce the robustness and repeatability of the key-generation method.

Response:

We appreciate the Reviewer #2's constructive comment regarding the stability of the evaluated figures of merit (FOM) under small processing variations. To address this, we have newly conducted a detailed sensitivity analysis of the PUF bitstream generation procedure. Specifically, we varied the threshold value by ±5% and ±10% during binarization, and additionally explored alternative binning sizes of 8×8 and 16×16, compared to the baseline 4×4 binning. For each configuration, we re-calculated key metrics including the average inter-HD, entropy, and uniformity. As shown in Figure R10, the results showed that the variations introduced by these changes were minimal, indicating strong robustness of the key-generation process. For instance, the variances of inter-HD, entropy, and uniformity across different threshold shifts remained low with confidence intervals (95%) for inter-HD, entropy, and uniformity were within $49.25 \pm 0.08\%$, $0.9916 \pm 0.0013\%$, and $50.02 \pm 1.93\%$, respectively, across the evaluated cases.

Similar, as represented in Figure R11, the proposed PA-PUF exhibited stable randomness performance regardless of the binning size variation. In this case, the specific confidence intervals (95%) for uniformity, entropy, and inter-HD were within $49.98 \pm 0.028\%$, 0.9855 ± 0.0098 , and $48.92 \pm 1.3\%$, respectively. Above-described characterization results suggest that the generated patterns and evaluated security metrics are stable against small shifts in preprocessing parameters.

We have included this additional analysis results and discussion in the revised manuscript as follows. Thank you very much for the Reviewer #2's helpful comments once again.

→ [R2Q3] To further evaluate the practical robustness of the PA-PUF key generation process, we performed additional sensitivity analyses to examine the effect of minor variations in preprocessing conditions. Specifically, we assessed how small shifts in binarization thresholds ($\pm 5\%$, $\pm 10\%$) and changes in binning sizes (4×4 , 8×8 , 16×16) affect the extracted bitstream quality. For each condition, we recalculated key figures of merit, including inter-HD, entropy, and uniformity. As shown in Supplementary Figs. 15 and 16, all metrics remained stable across the tested ranges. For instance, under threshold variation (Supplementary Fig. 15), the 95% confidence intervals for inter-HD, entropy, and uniformity were $49.25 \pm 0.08\%$, 0.9916 ± 0.0013 , and $50.02 \pm 1.93\%$, respectively. Similarly, under bin size variation (Supplementary Figs. 16), these values were $48.92 \pm 1.31\%$, 0.9855 ± 0.0098 , and $49.98 \pm 0.028\%$, indicating minimal degradation. These results suggest that the PA-PUF are resilient against minor perturbations in processing parameters, demonstrating the robustness and reliability of the system.

Figure R10 (Figure S15). Evaluation of PA-PUF randomness robustness under threshold shifts. (a) Binary response maps generated using five different binarization thresholds (-10%, -5%, 0%, +5%, +10%). (b) Average inter- HD under each threshold shift. (c) Average entropy values across threshold conditions, remaining close to the statistical ideal of 1. (d) Average uniformity results showing consistent bit balance near 50% even with $\pm 10\%$ threshold variation.

Figure R11 (Figure S16). Evaluation of randomness metrics of PA-PUF patterns under varying binning sizes. (a) Visual representation of PA-PUF binary patterns with different binning sizes (4, 8, and 16). (b) Uniformity distribution across bin sizes, showing near-ideal balance with 95% confidence interval (CI) of $49.98 \pm 0.028\%$. (c) Inter-HD distribution with a 95% CI of $48.92 \pm 1.31\%$, confirming randomness across devices. (d) Average entropy values across bin sizes, remaining close to the statistical ideal of 1 (CI = 0.9855 ± 0.0098).

R2Q4. The resistance evaluation is limited to logistic regression and SVM classifiers, yet advanced attackers often employ deep-learning techniques. Have the authors considered testing convolutional neural networks (CNNs) or transformer-based models to infer PA-PUF patterns and exploring adversarial training to probe vulnerabilities? Including at least one pilot experiment with a more complex neural architecture—or a brief rationale for why simpler models suffice—would significantly strengthen the manuscript’s security claims.

Response:

We sincerely appreciate the Reviewer #2 for giving us valuable and constructive suggestion. To strengthen the manuscript’s security claims, we have newly conducted an additional machine learning attack resilience evaluation using an advanced model, a convolutional neural network (CNN) architecture.

In this experiment, each challenge was derived from the (x, y) coordinate of a response bit and encoded into a 16-bit binary vector. This vector was reshaped into a two-dimensional format and used as a single-channel input to a CNN (Figure R12a). The CNN architecture includes convolutional and pooling layers for feature extraction, followed by ReLU activation and fully connected layers for binary response prediction. During training, the corresponding response bit (0 or 1) was provided as the ground truth label in a supervised learning setting. To evaluate learning behavior and attack performance, we varied the training dataset ratio from 10% to 70% and conducted five independent runs per setting. Prediction accuracy was averaged across runs to account for variability. As displayed in Figure R12b and c, The results showed that even at the largest training ratio (70%), the maximum prediction accuracy remained around 53.3%, which is close to random guessing (50%) with an average accuracy of 52.39%. This low predictability demonstrates the strong modeling resistance of the proposed PA-PUF, even against a CNN-based advance model attack. Through this additional evaluation, we further confirmed that the proposed PA-PUF exhibits robust resistance not only to classical machine learning attacks (logistic regression, SVM) but also to advanced modeling attacks (Figure R12d).

We have added the above experimental results and discussion in our revised manuscript as follows.

→ **[R2Q4]** To further validate the modeling resistance of the PA-PUF under more advanced inference scenarios, we additionally conducted an attack simulation using a convolutional neural network (CNN) model (Supplementary Fig. 18). In this evaluation, each response bit was treated as the ground truth label, and its corresponding challenge input was derived from the binary encoding of the (x, y) coordinates. The CNN architecture included convolutional and pooling layers with ReLU activation, followed by fully connected layers for binary classification. The model was trained in a supervised learning setting using varying training set sizes ranging from 10% to 70%, and five repeated trials were conducted to assess variability. As shown in Supplementary Figs. 18b-d, the CNN failed to exceed 53.3% prediction accuracy even at the highest training ratio, with an overall average accuracy of 52.39%. This result indicates that the proposed PA-PUF exhibits strong resistance not only against classical machine learning models, such as LR and SVM, but also against advanced neural network-based modeling attacks.

Figure R12 (Figure S18). Evaluation of PA-PUF robustness against machine learning-based modeling attacks. (a) Schematic illustration of the CNN-based attack model used to predict PUF responses. (b) Heatmap showing prediction accuracy of CNN across five trials with training data ratios ranging from 10% to 70%. (c) Distribution of prediction accuracy from CNN over multiple runs. (d) Comparison of prediction accuracy using logistic regression (LR), support vector machine (SVM), and convolutional neural network (CNN) across varying training data sizes.

R2Q5. Transfer onto human skin is demonstrated, but in-vivo conditions (skin temperature fluctuations, hydration changes, and movement) could introduce bit errors. How might these physiological factors influence the bit-error rate and overall reliability in a wearable application? Reflecting on these real-world influences and proposing mitigation strategies—such as adaptive thresholding or real-time calibration—would underscore the feasibility of practical wearable use.

Response:

We appreciate the Reviewer #2's thoughtful comments on the potential influence of physiological conditions in wearable settings. As the Reviewer #2 constructively commented, we have conducted additional experiments to investigate each of these factors individually, as detailed below.

(i) Temperature fluctuations:

To quantitatively evaluate the thermal robustness of our PUF system, we have conducted controlled heating experiments using a heating pad integrated into the sample stage, covering a physiologically relevant temperature range of 36.5 °C to 50 °C (Figure R6a). PA responses were collected at each temperature condition, and the resulting binary PUF patterns were analyzed using standard statistical metrics (Figure R6b–d).

Similarity analysis revealed that all pairwise comparisons between different temperatures maintained over 95% similarity (Figure R6c), indicating minimal pattern variation across thermal conditions. In addition, key figures of merit—including uniformity (49.1~49.18%), entropy (0.97~0.972), and inter-HD (49.8~49.9%)—remained close to ideal statistical values across all tested temperatures (Figure R6d). These results confirm that the thermal variation within the expected in-vivo range does not significantly affect the reproducibility or statistical properties of the PUF output.

(ii) Hydration levels:

Our system operates in a fully water-coupled configuration (e.g., via coupling gel or immersion), ensuring that localized hydration variability in skin tissue has negligible impact on signal amplitude or pattern recognition. This mitigates concerns regarding

signal instability due to moisture fluctuations.

(iii) Subject movement:

In the case of real-world physiological movement, we have conducted additional measurements to examine the impact of in-vivo motion on key stability. Specifically, four repeated PA measurements were performed with the PUF device attached to human skin under natural conditions, including respiration and subtle posture adjustments (Figure R13). The first acquisition was designated as the enrollment scan, while the subsequent three scans were treated as follow-up measurements. Pairwise comparison of the derived binary keys has revealed an average bit error rate of 10.35%, likely arising from motion-induced mechanical variability.

Following the Reviewer #2's comment to explore potential mitigation strategies, we have incorporated a fuzzy extractor into our system. A fuzzy extractor is a cryptographic primitive that enables reliable key regeneration from noisy physical sources such as biometrics and PUFs [R4-R6]. As illustrated in Figure R13e, during enrollment, it generates both a secure key and public helper data from the initial PUF response. In the reconstruction phase, this helper data allows the original key to be recovered from subsequent noisy inputs, provided the bit error remains within a tolerable bound. In our experiments, the fuzzy extractor has successfully reconstructed the original key with 100% reliability across all in-vivo measurements, despite the observed bit-level noise (Figure R13f). These results demonstrate that fuzzy extraction can serve as an effective post-processing strategy to ensure stable key regeneration under realistic wearable conditions, aligning well with the Reviewer #2's emphasis on practical applicability.

We have added a summary and experimental results discussing the PA-PUF system's robustness under physiological conditions. Once again, thank you very much for the Reviewer #2's constructive comments, which enhanced the clarity and completeness of our study.

→ [R2Q5] Thermal stability was also evaluated by heating the device to physiologically relevant temperatures (e.g., 36.5°C – 50 °C). The resulting PA responses exhibited consistent entropy and uniformity values, and the bit-level similarity remained above 95% across conditions (Supplementary Fig. 22).

Revised Manuscript (PAGE 15)

→ [R2Q5] To further evaluate the influence of in-vivo physiological motion—such as respiration, heartbeat, and minor posture shifts—on PUF stability, we have conducted repeated measurements while the device was attached to human skin as illustrated in Supplementary Fig. 23. Four PA images were acquired over time, with the first defined as the enrollment scan and the remaining three as subsequent scans. Despite natural subject movement, the resulting PUF responses exhibited consistent bit-level similarity, with an average bit error rate of 10.35%. Key statistical metrics (uniformity, entropy, and inter-HD) remained close to ideal values across all scans. To mitigate this level of variability and ensure robust key regeneration, we have incorporated a fuzzy extractor—a cryptographic primitive designed to generate stable keys from noisy physical inputs^{39, 40, 41}. During the enrollment phase, a key and associated helper data are derived from the initial PUF output. In subsequent authentication phases, the helper data enables recovery of the original key from noisy PUF responses, provided the bit error remains within a correctable range. This process ensures that the extracted key remains consistent even under in-vivo fluctuations, without compromising security. Using this approach, we have successfully reconstructed the original cryptographic key from all subsequent scans with 100% accuracy. These results demonstrate the effectiveness of fuzzy extraction in compensating for motion-induced noise, reinforcing the practicality of the proposed PA-PUF system for secure wearable applications.

[R4] Mahendran, R. K., & Velusamy, P. (2020). A secure fuzzy extractor based biometric key authentication scheme for body sensor network in Internet of Medical Things. *Computer Communications*, 153, 545-552.

[R5] Arenas, M. P., Fotiadis, G., Lenzini, G., & Rakeei, M. (2025). Remote secure object authentication: Secure sketches, fuzzy extractors, and security protocols. *Computers & Security*, 148, 104131.

[R6] Chandran, I., & Vipin, K. (2024). A PUF secured lightweight mutual authentication protocol for Multi-UAV networks. *Computer Networks*, 253, 110717.

Figure R6 (Figure S22). (a) Schematic illustration of the experimental setup for thermal stability testing. (b) Digitized 250×250 PA PUF patterns extracted from each PA MAP image. (c) Similarity analysis of each PA response. A single PUF device was relatedly tested at the same region. (d) Uniformity distribution, entropy analysis, and inter-HD distribution of each PA PUF responses.

Figure R13 (Figure S23). Evaluation of PUF key stability under in-vivo motion conditions and recovery using a fuzzy extractor. (a) Schematic representation of repeated PA pattern acquisition with the PUF device attached to human skin under in-vivo motion conditions. (b) Digitized PA PUF patterns extracted from each PA image scans. The first scan is defined as the enrollment scan, followed by three subsequent scans taken at different time points under natural motion (e.g., respiration, subtle posture adjustments). (c) Similarity matrix showing pairwise bit-level comparisons among the four scans. An average bit error rate of 10.35% was observed between the enrollment and subsequent scans. (d) Uniformity distribution, entropy analysis, and inter-HD distribution of each PA PUF responses. (e) Conceptual diagram of fuzzy extractor implementation. During enrollment, a key and helper data are generated; in the reproduction phase, noisy inputs are corrected using the stored helper data to regenerate the same key. (f) Key similarity results before and after applying the fuzzy extractor.

R2Q6. Ambient acoustic noise in non-laboratory settings (e.g., machinery, human activity) could corrupt PA signals and key extraction. Have the authors characterized the system's signal-to-noise ratio under controlled noise sources and implemented filtering or time-gating techniques to reject background interference? Demonstrating noise-immunity thresholds through such tests would convincingly illustrate the approach's practicality outside idealized environments.

Response:

We appreciate the Reviewer #2's detailed review and constructive comments. In PA imaging, the detected signals lie in the ultrasonic frequency range, not the audible range (~Hz–kHz). Our system employs a transducer with a center frequency of 25 MHz, and we collect signals within the effective detection bandwidth of approximately 5–30 MHz. As a result, ambient acoustic noise from human activity or machinery, which typically exists below 20 kHz, does not spectrally overlap with the PA signals used in our system. This frequency separation ensures that environmental noise does not interfere with the reliable acquisition of PA responses.

To validate the robustness against ambient acoustic noise, we conducted additional experiments and analyzed the stability of PUF characteristics before and after introducing controlled ambient noise conditions (Figure R14). For each configuration, we calculated key metrics including the inter-HD, entropy, uniformity, and similarity. The inter-HD, entropy, and uniformity across the ambient noise stability tests remained high at approximately 49.84%, 0.984, and 50.15%, respectively. Furthermore, the similarity remained above 95.04% for PA responses, confirming the system's resilience under realistic acoustic environments.

However, we acknowledge that other types of noise such as thermal noise, power-line interference, and amplifier noise can potentially degrade the signal-to-noise ratio (SNR) in practical settings. To address this, we implemented a bandpass filter specifically matched to the bandwidth of our ultrasound transducer (5–30 MHz). This filtering approach helps suppress both low-frequency and high-frequency noise components that fall outside the signal band. To confirm its effectiveness, we measured the SNR before and after filtering and observed a clear improvement from 19.06 dB to 25.46 dB (Figure R15), supporting the validity of our noise reduction strategy.

We sincerely thank the Reviewer #2 for this insightful and helpful comment. We have added the above experimental results and corresponding discussion in our revised manuscript as follows.

Revised Manuscript (PAGE 15-16)

→ [R2Q6] Finally, to evaluate ambient noise resilience, the system was tested under controlled background noise. Since the PA signals lie within the 5–30 MHz range and ambient noise sources typically fall below 20 kHz, spectral overlap is negligible. Furthermore, by applying a bandpass filter matched to the transducer bandwidth, we improved the signal-to-noise ratio (SNR) from 19.06 dB to 25.46 dB (Supplementary Fig. 24), and the resulting cryptographic patterns remained highly consistent (similarity > 95.04%) (Supplementary Fig. 25). These findings confirm that the proposed PA-PUF exhibits strong resilience to thermal, mechanical, acoustic, and physiological perturbations, highlighting its practical potential for secure and robust wearable applications.

Figure R14 (Figure S25). Stability of PA-PUF responses under ambient acoustic noise. (a) Scheme of the measurement setup for ambient noise stability test. (b) Digitized 250×250 PA PUF patterns extracted from the PUF device before and after gaining the ambient noise test. (c) Inter-HD distribution of each PA PUF response. (d) Entropy analysis of each PA PUF response. (e) Uniformity distribution of each PA PUF responses. A single PUF device was relatedly tested at the same region.

Figure R15 (Figure S24). Noise reduction in PA imaging using bandpass filtering. Comparison of raw and filtered photoacoustic images. The red boxes indicate regions of interest used for signal and the yellow boxes indicate regions of interest used for noise, demonstrating improved signal clarity after noise filtering.

R2Q7. The PA response intensity strongly depends on local film thickness and nanoparticle dispersion, so uneven coatings could bias bit distributions. Can the authors provide profilometry, ellipsometry, or cross-sectional SEM data to verify thickness uniformity across the 2.5×2.5 cm² substrate and quantify acceptable non-uniformity tolerances? Outlining coating-optimization strategies (e.g., optimized spin- versus flow-coating parameters) would substantiate the method’s scalability and ensure consistent key generation over large areas.

Response:

We greatly appreciate the Reviewer #2’s considerable and valuable comments. As the Reviewer #2 constructively suggested, we have newly conducted a thorough evaluation of film thickness uniformity across the entire 2.5 × 2.5 cm² substrate by using both profilometry and cross-sectional SEM measurements. As shown in Figure R16a, nine representative regions (i–ix) were selected from across the substrate area. The results from profilometry (Figure 16b–d) demonstrate consistent film height along 100 μm line scans, with no abrupt deviations or local discontinuities. The corresponding median thickness values, summarized in the color map, range from 129.3 nm to 152.9 nm, yielding an overall average of 140.31 nm and a standard deviation of ±10.05 nm. This variation corresponds to less than ±10% deviation from the mean, indicating a high degree of thickness uniformity. These findings are corroborated by cross-sectional SEM images (Figure R17), which independently confirm the measured thicknesses across the same regions.

Regarding non-uniformity tolerance, we would like to offer a perspective from the PA imaging standpoint. Our PA system has a lateral resolution of approximately 1 μm, while the optical depth of field (DOF) extends over several hundred micrometers. This depth range is much larger than the sub-micrometer thickness of our nanoparticle film. As a result, the PA signal effectively integrates over the entire film thickness, making it inherently insensitive to small vertical variations such as minor thickness fluctuations within ±10%. In practice, we found that the PA response is more strongly affected by in-plane non-uniformities, such as lateral aggregation or the presence of voids in the film. Thus, we believe that the small thickness variations observed in our device are unlikely to introduce any significant bias in the bit generation process.

From the perspective of improving coating uniformity and scalability, several practical strategies can be considered beyond conventional spin coating. Techniques such as flow

coating or slot-die coating are particularly promising, as they can provide more uniform shear forces and better control of solvent evaporation across large areas, which is beneficial for achieving consistent film quality. In addition, tuning the physical properties of the coating solution, such as increasing viscosity or optimizing surface tension, can help minimize dewetting and encourage uniform spreading during deposition. The surface of the substrate also plays an important role. Treatments such as plasma activation or UV-ozone exposure can enhance nanoparticle adhesion and improve overall coating uniformity. Finally, integrating real-time monitoring tools such as interferometry or optical reflectometry into the coating process can enable dynamic feedback and correction, helping to ensure reliable and reproducible film thickness across the substrate.

We sincerely thank the Reviewer #2 for the valuable suggestion. We have added the corresponding experimental results and discussions in our revised manuscript as follows.

Revised Manuscript (PAGE 8)

→ [R2Q7] To quantitatively evaluate the thickness uniformity of the CuO/SnO₂ composite film across the full deposition area ($2.5 \times 2.5 \text{ cm}^2$), we conducted profilometry and cross-sectional SEM measurements over nine representative regions distributed throughout the substrate (Supplementary Fig. 4). Line-scan profilometry revealed consistent height profiles over 100 μm distances with no abrupt changes. The median thickness values extracted from each region ranged from 129.3 nm to 152.1 nm, with an average of 140.31 nm and a standard deviation of $\pm 10.05 \text{ nm}$. This corresponds to less than $\pm 10\%$ deviation from the mean thickness, indicating high spatial uniformity across the film. Cross-sectional SEM images obtained at the same regions (Supplementary Fig. 5) independently confirmed this uniformity, with measured thicknesses closely matching the profilometry results.

Figure R16 (Figure S4). Surface profile characterization of the PA-PUF film. (a) Optical image of the PA-PUF sample showing nine distinct regions marked for line-scan analysis. (b) Surface height profiles obtained from each region indicated in (a), labeled (i) to (ix). (c) Median surface height values for each region. (d) Statistical distribution of surface heights with a mean of 140.31 nm and a standard deviation of 10.05 nm.

Figure R17 (Figure S5). Cross-sectional SEM images of the PA-PUF film. Thickness measurements from nine different regions, corresponding to the regions shown in Figure R16a. The measured average film thickness for each region range from 125.8 nm to 154.1 nm, confirming uniformity across the scanned area. Scale bar represents 500 nm.

We very much appreciate the Reviewers' all the constructive and valuable comments, which greatly helped us to improve this manuscript.